# TempCLR: Temporal Alignment Representation with Contrastive Learning

**Yuncong Yang**[†]    **Jiawei Ma**[†]    **Shiyuan Huang**    **Long Chen**    **Xudong Lin**
**Guangxing Han**    **Shih-Fu Chang**
Columbia University, New York, NY 10027, USA
{yy3035,jiawei.m,shiyuan.h,cl3695,xudong.lin,gh2561,sc250}@columbia.edu

## Abstract

Video representation learning has been successful in video-text pre-training for zero-shot transfer, where each sentence is trained to be close to the paired video clips in a common feature space. For long videos, given a paragraph of description where the sentences describe different segments of the video, by matching all sentence-clip pairs, the paragraph and the full video are aligned implicitly. However, such unit-level comparison may ignore global temporal context, which inevitably limits the generalization ability. In this paper, we propose a contrastive learning framework TempCLR to compare the full video and the paragraph explicitly. As the video/paragraph is formulated as a sequence of clips/sentences, under the constraint of their temporal order, we use dynamic time warping to compute the minimum cumulative cost over sentence-clip pairs as the sequence-level distance. To explore the temporal dynamics, we break the consistency of temporal succession by shuffling video clips w.r.t. temporal granularity. Then, we obtain the representations for clips/sentences, which perceive the temporal information and thus facilitate the sequence alignment. In addition to pre-training on the video and paragraph, our approach can also generalize on the matching between video instances. We evaluate our approach on video retrieval, action step localization, and few-shot action recognition, and achieve consistent performance gain over all three tasks. Detailed ablation studies are provided to justify the approach design.

## 1 Introduction

Representation learning on videos has achieved success (Goroshin et al., 2015; Feichtenhofer et al., 2021) in detecting actions in short periods. Recent work has extended it on video-text data (Miech et al., 2019; Radford et al., 2021) to learn a common feature space for zero-shot transfer. In particular, given a paragraph of description, the understanding of long videos is increasingly important and may facilitate AI-assistant applications (Grauman et al., 2022; Lin et al., 2022; Chen et al., 2022).

A long video is usually formulated as a sequence of short video clips. Given a paragraph, each sentence is used to describe, *i.e.*, paired with, the consecutive video clips in a video segment. By matching all sentence-clip pairs (Miech et al., 2020), the full video and the paragraph can be aligned implicitly. However, maximizing the agreement between clips and sentences individually (*unit-level*) ignores the context of temporal dynamics, which limits the generalization (Goyal et al., 2017). After all, within one video segment, as the action/event progress at each clip varies, the similarity between the clips and the sentence can be naturally different. As such, strictly aligning the sentence with all paired clips, serving as the hard-label, may not always result in an optimal solution.

To incorporate the temporal correlation across clips, Xu et al. (2021) propose to first fuse the representations over a short period for sentences and video clips separately and then align the fused representations. However, such methods only incorporate the *local* temporal information but still does not model the *global* temporal correlation. As a paragraph is essentially a sequence of sentences, as shown in Fig. 1, the whole long video and the paragraph should be explicitly compared and aligned (*sequence-level*). For a video consisting of multiple steps, *e.g.*, instructional video, the

---

[†]Equal contribution. Code Link: https://github.com/yyuncong/TempCLR

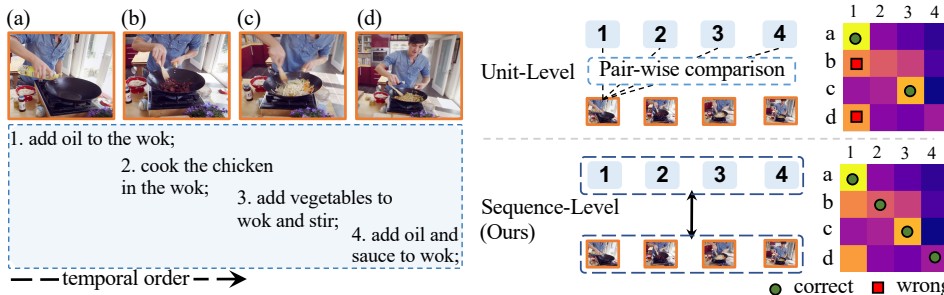

Figure 1: (Left) Given a video and a paired paragraph where the sentences can describe the content in different parts, the temporal orders within the video and the paragraph are consistent. (Right) Conventional methods perform unit-level comparison between sentences and clips pair-wisely and mismatch may occur due to visual similarity. Instead, we directly compare the sequences by considering temporal order such that the temporal succession can be used to align clips and captions.

temporal dependence between two distant video clips still exists. In this way, for a challenging case where two clips are visually similar but are from different segments (clips $\{a, b, d\}$ in Fig. 1), the global context of order can be utilized to avoid the potential mismatching in unit-level matching.

In this work, we study video-paragraph pre-training and propose a framework TempCLR based on sequence-level comparison to explore temporal dynamics. We directly calculate the distance between full video and paragraph. Without loss of generality, for the paragraph (anchor) and its paired video (positive sample), the sequence-level distance is the minimum cumulative matching cost over the sentences and clips under the constraint of temporal order and is obtained via dynamic time warping (DTW) (Müller, 2007). Then we emphasize the unit order which is naturally exhibited within a sequence, and consider the cases where the temporal consistency between video and paragraph is not met. As a sentence is paired with a segment consisting of multiple clips, we design a negative sampling strategy based on temporal granularity, which shuffles the clips at both unit level and segment level in the paired video. Finally, we apply contrastive learning to maximally align paired video and paragraph. In this way, we can learn representations for clips and sentences which can perceive global temporal context. Then, from the optimal matching with minimum sequence-level distance, we can pair clips with sentences without being confused by visual similarity.

In addition to video-paragraph pre-training, our TempCLR can also be generalized on few-shot action recognition (video-only) where each video is classified through the nearest neighbor search according to the sequence distance. In summary, the contributions are:

- We propose a contrastive learning framework TempCLR to explore temporal dynamics where the learned representation for clips and sentences can facilitate the alignment between sequences.

- Given an anchor, we design a negative sampling strategy based on temporal granularity and shuffle the units in the positive sequence at both segment-level and unit-level. Notably, our method can be generalized to learn representation for both video-paragraph data and video-only data.

- We conduct extensive experiments on three tasks (*i.e.*, video retrieval, action step localization, and few-shot action recognition) and achieve consistent performance gain to demonstrate the effect of our training strategy. Detailed ablation studies are provided to justify the approach design.

## 2 RELATED WORK

**Contrastive learning** (CT) has achieved success on images (Chen et al., 2020b; He et al., 2020) and can cluster samples in the feature space properly. The main idea is to group different views of the same image/video instance by minimizing the InfoNCE loss (Oord et al., 2018) while Wang & Isola (2020) explains it from the aspect of uniformity. Besides, it can be applied on imbalanced dataset (Caron et al., 2020) and different feature spaces (Grill et al., 2020; Ma et al., 2021).

**Self-supervision on Video** has been studied to learn a good feature extractor. Following the idea of temporal jigsaw Huo et al. (2020), the order of frames (Misra et al., 2016; Lee et al., 2017) and

clips (Xu et al., 2019) can be used for supervision. However, the length of tuple is fixed and too small (*e.g.*, 4) to model temporal context over the full video and may ignore semantic information. Besides, contrastive learning is also applied for video pre-training where the positive samples can be built by randomly selecting shot video clips (Feichtenhofer et al., 2021), adjusting the time span (Recasens et al., 2021), and performing spatial-temporal augmentation (Jenni et al., 2020). In addition, generative approaches Han et al. (2020); Vondrick et al. (2016) has also been studied.

**Multi-modal pre-training for zero-shot transfer** has been studied to connect vision and language. CLIP (Radford et al., 2021) applies contrastive learning on image-caption pairs. Yang et al. (2022) and Li et al. (2022) further modify the negative sampling strategy such that the embeddings can be more discriminative at instance-level. Then the pre-training is extended to video understanding Miech et al. (2020); Ging et al. (2020); Gabeur et al. (2020); Alayrac et al. (2020); Wang et al. (2022). To improve the performance, multi-task training has been studied (Li et al., 2020; Luo et al., 2020). As the description of video content can be noisy, TAN (Han et al., 2022) proposes a co-training strategy and use mutual agreement for annotation filtering and VideoCLIP (Xu et al., 2021) proposes a sampling strategy to mitigate the impact of noise in long videos labeling. Besides video and text, audio is also used to benefit the zero-shot tasks (Chen et al., 2021; Shvetsova et al., 2022) by learning a common feature space. As an alternative, the Attention mechanism can also be used to fuse the multi-modal information at each layer (Sun et al., 2019; Su et al., 2019; Chen et al., 2020c).

**Sequence alignment** For each unit in a sequence, along the temporal order, the indices of matched units from aligned sequence pair shall monotonically increase, and averaged distance over matched units is minimized. Dynamic time wrapping (Müller, 2007) is first proposed and canonical time warping (Zhou & Torre, 2009) is then used to align sequences with different feature dimensionality and is applied in deep learning (Sargin et al., 2007). Meanwhile, a pyramid deep architecture (Wang et al., 2020) or attention mechanism (Bishay et al., 2019; Zhang et al., 2020; Ma et al., 2019) can be designed to integrate multi-scale temporal information into a single feature vector. Besides, regularized by the sequence alignment, pre-training strategies are designed for visual-audio/rhythms synchronization (Cheng et al., 2020; Yu et al., 2022), and video-text alignment (Xu et al., 2021).

## 3 APPROACH

We first provide notation and task formulation in Sec. 3.1. Then, we detail the paradigm of our method in Sec. 3.2 and explain how to adapt it for different tasks in 3.3.

### 3.1 PRE-TRAINING TASK FORMULATION

Given an anchor instance $\mathbf{S}_a$ (*i.e.*, a paragraph or a video), we aim to learn a network that can minimize its distance with a positive instance $\mathbf{S}_p$ (*i.e.*, a video paired with the paragraph or a video of the same semantic class). For each paragraph/video, since its time span can be long, it is typically formulated as a sequence of sentences/video clips. Then, a network is trained to extract a feature for each sentence/video clip, resulting in a sequence of feature embeddings, *i.e.*, $\mathbf{S}_a = \{\mathbf{s}_a^i\}_{i=1}^{N_a}$ and $\mathbf{S}_p = \{\mathbf{s}_p^j\}_{j=1}^{N_p}$, where $\mathbf{s}_a^i, \mathbf{s}_p^j \in \mathcal{R}^d$ are the sequence units, $N_a$ and $N_p$ are the sequence lengths, and $d$ is the dimension of the common feature space. In a pair of video and paragraph, the sentence can be mapped with a few consecutive clips, *i.e.*, $\mathbf{s}_a^i$ is matched with $\{\mathbf{s}_p^j\}_{j=t_i^0}^{t_i^1}$ where $\{t_i^0, t_i^1\}$ are the starting and ending feature indexes in $\mathbf{S}_p$ for $\mathbf{s}_a^i$. Then, $N_a$ is not necessarily equal to $N_p$. In this way, as the intrinsic temporal orders within $\mathbf{S}_a$ and $\mathbf{S}_p$ are consistent, their distance $d_{\{\mathbf{S}_a, \mathbf{S}_p\}}$ should be small. In contrast, two sequences should be distant from each other if they cannot be aligned.

### 3.2 TEMPORAL ALIGNMENT REPRESENTATION WITH CONTRASTIVE LEARNING

In this section, we explain our contrastive(CT)-based training framework TempCLR. We consider the sequences consisting successive steps of a common topic. Then, we propose our negative sampling strategy and choose to use sequence-level comparison to directly calculate the distance.

The temporal dynamics exhibited in each sequence (*i.e.*, paragraph or video) are representative. To align two sequences, it is also important to ensure the units within the sequences are properly matched. For example, as $\mathbf{S}_a$ and $\mathbf{S}_p$ are of temporal consistency, when $\mathbf{S}_a$ and $\mathbf{S}_p$ are aligned, each

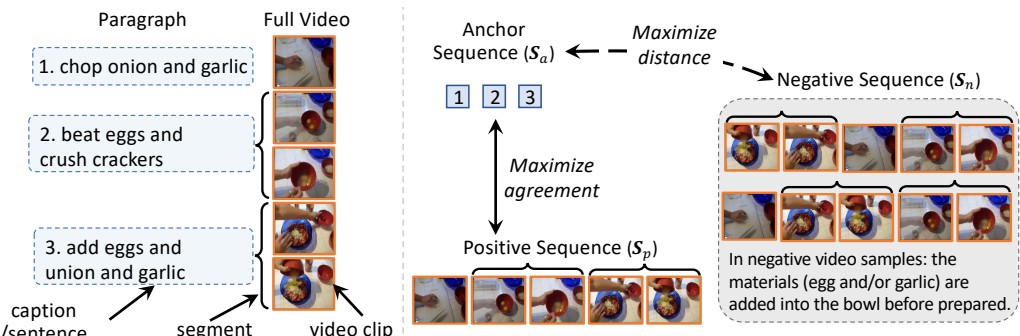

Figure 2: (Left) For each video-paragraph pair, each sentence describes one segment consisting of consecutive video clips. (Right) We set the paragraph (*i.e.*, sequence of sentences) as anchor $\mathbf{S}_a$ and set the paired video (*i.e.*, sequence of clips) as positive sample $\mathbf{S}_p$. Then, we shuffle the units in $\mathbf{S}_p$ to generate a negative sequence $\mathbf{S}_n$ as the temporal succession in $\mathbf{S}_n$ is not consistent with $\mathbf{S}_a$. For clip shuffling, we first alter segments and then optionally shuffle the clips in each segment.

unit $\mathbf{s}_a^i$ in $\mathbf{S}_a$ should also be matched with $\mathbf{s}_p^j$ in $\mathbf{S}_p$ where $\mathbf{s}_a^i$ and $\mathbf{s}_p^j$ are assumed to be semantically close to each other. In this way, when two features $(\mathbf{s}_p^i, \mathbf{s}_p^j)$ in $\mathbf{S}_p$ are of two different actions but hard to be distinguished due to visual similarity, the network can still utilize the global temporal order to find a proper unit-level matching between $\mathbf{S}_a$ and $\mathbf{S}_p$ (shown in Fig. 1).

**Negative Sampling based on Temporal Granularity.** In contrast, for a negative sequence $\mathbf{S}_n = \{\mathbf{s}_n^j\}_{j=1}^{N_n}$ which does not preserve temporal consistency with $\mathbf{S}_a$ where $N_n$ is the sequence length, the distance $d_{\{\mathbf{S}_a, \mathbf{S}_n\}}$ between $\mathbf{S}_a$ and $\mathbf{S}_n$ should be high. After all, when $\mathbf{S}_a$ cannot be aligned with $\mathbf{S}_n$, considering the global temporal order, each unit $\mathbf{s}_n^j$ is also distant from any units $\mathbf{s}_a^i$ in $\mathbf{S}_a$ though $\mathbf{s}_a^i$ and $\mathbf{s}_n^j$ can be visually similar (*e.g.*, similar background or foreground objects).

As such, to mitigate the impact from spatial information, we consider preserving temporal order and aim to obtain representations that facilitates the alignment. In detail, as shown in Fig. 2, we propose to generate negative sequence $\mathbf{S}_n$ by randomly shuffling $\mathbf{S}_p$ and breaking temporal consistency. Since the segment-level matching also exists in the paired sequences, we first alter the segment order and then shuffle the units within each segment. To effectively compare $\mathbf{S}_n$ and $\mathbf{S}_p$ with $\mathbf{S}_a$, we propose a training framework TempCLR based on contrastive learning.

**Contrastive learning** (CT) is a self-supervision approach by learning to group the samples with high correlation. CT treats different views of the same instance as correlated (positive) and builds negative pairs by sampling views from different instances. As each instance is an image in Chen et al. (2020a), each view can be generated by performing augmentation on the low-level pixels, *i.e.*, adding effects such as flipping and color distortion. In practice, given a set $\mathcal{B}$ with $N_B$ instances, for each instance with one single view $I \in \mathcal{B}$, another view $I'$ is generated and is then used to build a positive pair with $I$ where the view of other instances in $\mathcal{B}$ are used to build negative pairs with $I$. Then, the training objective is to minimize the InfoNCE loss (Oord et al., 2018),

$$\mathcal{L}_{CT}(I, I', \mathcal{B}_n) = -\log \frac{\exp(\mathbf{z}_I \cdot \mathbf{z}_{I'}/\tau)}{\exp(\mathbf{z}_I \cdot \mathbf{z}_{I'}/\tau) + \sum_{X \in \mathcal{B}_n} \exp(\mathbf{z}_I \cdot \mathbf{z}_X/\tau)}. \tag{1}$$

where $\mathcal{B}_n = \mathcal{B} \setminus \{I\}$, $\mathbf{z}_X \in \mathcal{R}^d$ is the feature for instance $X$ after $l_2$-normalization and $\tau$ is a hyperparameter to rescale the affinity scores. In this way, CT is performing pair-wise comparison where the disagreement between $\mathbf{z}_I$ and $\mathbf{z}_{I'}$ is induced by the variation of augmentation.

Note, for our approach TempCLR, $\mathbf{S}_a$ and $\mathbf{S}_p$ served as *two different views* and the *pattern of temporal order that is consistent between* $\mathbf{S}_a$ *and* $\mathbf{S}_p$ *is treated as the instance from which the views are augmented*. Then, we can derive the training objective, *i.e.*,

$$\mathcal{L}_{seq}(\mathbf{S}_a, \mathbf{S}_p, \mathcal{S}_n) = -\log \frac{\exp(d_{\{\mathbf{S}_a, \mathbf{s}_p\}}/\tau)}{exp(d_{\{\mathbf{S}_a, \mathbf{s}_p\}}/\tau) + \sum_{\mathbf{S}_n \in \mathcal{S}_n} \exp(d_{\{\mathbf{S}_a, \mathbf{s}_n\}}/\tau)} \tag{2}$$

where $\mathcal{S}_n = \{\mathbf{S}_n^{(i)}\}_{i=1}^N$ is the set of $N$ negative sequences derived from $\mathbf{S}_p$. As a complementary component, the sequences generated from other instances unpaired or uncorrelated with $\mathbf{S}_a$ can

also be included in $\mathcal{S}_n$. However, it introduces more computation workload but does not improve performance effectively (analyzed in Sec. 5.2). As such, for each $\mathbf{S}_a$, we only use the shuffling strategy for negative sample generation. By minimizing $\mathcal{L}_{seq}$, $\mathbf{S}_a$ and $\mathbf{S}_p$ are trained to be close to each other while both of them are pushed away from all negative sequences in $\mathcal{S}_n$. Meanwhile, comparing with unit-level comparison between captions and clips, our approach provides a soft supervision on unit-level comparison and we provide detailed comparison in Sec. A.4.6.

**Sequence-level distance.** To jointly consider the unit-level matching in sequence alignment, we choose to use Dynamic Time Wrapping (DTW, Müller (2007)), which calculates the minimum cumulative matching costs over units as the sequence distance. The set of matched units is called optimal alignment and is recorded in a matching matrix $M \in \mathcal{R}^{N_a \times N_p}$ with binary indicators.

There are two constraints in DTW, 1) $\forall \mathbf{s}_p^j \in \mathbf{S}_p$ is supposed to be matched with at least one unit in $\mathbf{S}_a$, *i.e.*, $\sum_{1 \leq i \leq N_a} M(i,j) \geq 1$ for $\forall j \in \{1...N_p\}$, and vice versa; 2) Following the exhibited temporal order, the index of matched units should be monotonously increasing, *i.e.*, if $\mathbf{s}_p^n$ matches $\mathbf{s}_a^m$ where $1 \leq m \leq N_a$ and $1 \leq n < N_p$, $\mathbf{s}_p^{n+1}$ cannot match with any unit in $\{\mathbf{s}_a^i\}_{i=1}^{m-1}$ and $\sum_{1 \leq i < m} M(i,j) = 0$. Thus, for $\mathcal{S}_a$ and $\mathcal{S}_p$, when $M(i,j) = 1$, $\mathbf{s}_a^i$ and $\mathbf{s}_p^j$ are matched and assumed to be semantically close to each other. For implementation, we first calculate the pair-wise matching costs $D \in \mathcal{R}^{N_a \times N_p}$ where $D(i,j)$ is the cost, *i.e.*, cosine distance (Singhal et al., 2001), between $\mathbf{s}_a^i$ and $\mathbf{s}_p^j$. Then, DTW employs dynamic programming and sets a matrix $C \in \mathcal{R}^{N_a \times N_p}$ to record the minimum cumulative cost between $\{\mathbf{s}_a^i\}_{i=1}^{n_a}$ and $\{\mathbf{s}_p^j\}_{j=1}^{n_p}$ (Wang et al., 2020; Dixit et al., 1990), *i.e.*,

$$C(n_a, n_p) = D(n_a, n_p) + \min\{C(n_a - 1, n_p - 1), \ C(n_a - 1, n_p), \ C(n_a, n_p - 1)\}. \quad (3)$$

where $1 \leq n_p \leq N_p$ and $1 \leq n_a \leq N_a$. Then, the sequence distance is $d_{\{\mathbf{S}_a, \mathbf{S}_p\}} = C(N_a, N_p)$. Besides, Cao et al. (2020) propose OTAM, a variant of DTW, to avoid restrict boundary constraint DTW (Müller, 2007). The effect of DTW and OTAM in TempCLR are also studied in Sec. 4 & 5.

### 3.3 Adaptation for Pretraining Tasks

We briefly explain how to apply the paradigm to align the video and paragraph in one semantically-correlated pair (video-paragraph) or videos of the same class (video-only) during network training.

**Video-Paragraph.** For each long video, a paired paragraph consisting of short sentences is provided and every sentence, *i.e.*, caption, describes the visual content within a temporal segment. Firstly, for paragraph, we learn a text encoder to extract feature $\mathbf{s}_a^i$ for each sentence. Then, for the video, all frames are grouped into non-overlapping consecutive video clips and each clip, serving as a visual token, has $n_f$ frames. Thus, each sentence is paired with multiple consecutive clips. We use a backbone to extract token embedding for each clip. Then, a video encoder is trained to map each clip embedding to the clip feature where all clip features for all sentences are concatenated as $\mathbf{S}_p$. As the segments for different sentences may have overlap, during training, we will sub-select the sentences to have $\mathbf{S}_a$ such that there is no repeating clips in $\mathbf{S}_p$.

**Video-only.** Since the duration of action is usually short, we formulate the video as the sequence of frames. As there is no temporal segmentation within each action, we generate the negative sequences by directly shuffling the frames. In this way, the network is learned in a self-supervised manner.

## 4 Experiment

We first conduct experiments on video retrieval and action step localization to explain the benefit of TempCLR for zero-shot transfer in long video understanding. Then, we explain the experiments on few-shot action recognition to indicate the generalization ability from global temporal modelling on new classes. For the convenience of description, we interchangeably use caption and sentence.

### 4.1 Implementation Details of Video-Text Pre-Training

We follow Xu et al. (2021) and use HowTo100M (HT100M) (Miech et al., 2019) for pre-training. As HT100M is too large ( 1.2M videos), due to limited computation resource, we build our model on top of the VideoCLIP (Xu et al., 2021) *i.e.*, initialize the VideoCLIP network with its fully-trained model, and randomly select 90k videos (7.5%) of HT100M to update the network by minimizing

Table 1: Performance (%) comparison on full-video retrieval.

| Exp. | (*Background Removed*) | Measure | R@1 | R@5 | R@10 |
|---|---|---|---|---|---|
| 1 | MIL-NCE* | Cap. Avg. | 43.1 | 68.6 | 79.1 |
| 2 | HT100M* | Cap. Avg. | 46.6 | 74.3 | 83.7 |
| 3 | MCN (Chen et al., 2021) | Cap. Avg. | 53.4 | 75.0 | 81.4 |
| 4 | VideoCLIP[†] | Cap. Avg. | 74.5 | 94.5 | 97.9 |
| 5 | VideoCLIP[†] | DTW | 56.0 | 89.9 | 96.3 |
| 6 | TempCLR(Ours) | Cap. Avg. | 74.5 | 94.6 | 97.0 |
| 7 | TempCLR(Ours) | DTW | **83.5** | **97.2** | **99.3** |
| | (*Background Kept*) | Measure | R@1 | R@5 | R@10 |
| 8 | VideoCLIP[†] | DTW | 55.7 | 93.1 | **98.9** |
| 9 | TempCLR | DTW | **70.4** | **93.8** | 97.9 |

*:reported in Chen et al. (2021), [†]: our implementation

Table 2: Ablation Study on full-video retrieval (%)

| *Background Removed* | |
|---|---|
| Measure | TempCLR |
| DTW | **83.5** |
| OTAM | 83.4 |
| *Background Kept* | |
| Measure | VideoCLIP |
| DTW | 55.7 |
| OTAM | 53.9 |
| Measure | TempCLR |
| DTW | 70.4 |
| OTAM | **72.2** |

(Metric: R@1 × 100%)

Table 3: Performance (%) comparison on video retrieval (clip-caption)

| Approach | backbone | R@1 | R@5 | R@10 |
|---|---|---|---|---|
| Random | - | 0.0 | 0.2 | 0.3 |
| MIL-NCE*(Miech et al., 2020) | R152+RX101 | 8.1 | 23.3 | 32.3 |
| MCN(Chen et al., 2021) | R152+RX101 | 18.1 | 35.5 | 45.2 |
| MMV(Alayrac et al., 2020) | TSM-50x2 | 11.7 | 33.4 | 45.4 |
| ActBERT(Zhu & Yang, 2020) | R101+Res3D | 9.6 | 26.7 | 38.0 |
| MIL-NCE(Miech et al., 2020) | I3D-G | 11.4 | 30.6 | 42.0 |
| HT100M(Miech et al., 2019) | S3D-G | 6.1 | 17.3 | 24.8 |
| MIL-NCE(Miech et al., 2020) | S3D-G | 15.1 | 38.0 | 51.2 |
| MMFT (Shvetsova et al., 2022) | S3D-G | **24.6** | 48.3 | 60.4 |
| VideoCLIP (Xu et al., 2021) | S3D-G | 22.7 | 50.4 | 63.1 |
| TempCLR(Ours) | S3D-G | 23.3 | **51.0** | **64.5** |

$\mathcal{L}_{seq}$. VideoCLIP consists of two Transformers (Vaswani et al., 2017) as encoders for video and paragraph separately. For each clip, they use its S3D feature (Xie et al., 2018) as embedding. For each sentence, the token embeddings are obtained via a lookup table (Devlin et al., 2018). One MLP layer is set to map clip embeddings and align the dimension of sentence embeddings. During pre-training, all encoder parameters are updated. More experiment details can be found in Appendix.

## 4.2 VIDEO-TEXT DOWNSTREAM EVALUATION

### 4.2.1 VIDEO RETRIEVAL

**Setup and Metric.** We evaluate our approach under two settings *Full-Video* and *Caption-Clip*. (*Full-Video*) Given a paragraph which contains a set of sentence queries, describing multiple parts of an entire long video, the full video should be directly retrieved. To represent the full video, we can either concatenate all clips which have paired captions (*i.e.*, remove background), or directly use the full video with background. For retrieval, we can use DTW to measure the distance between the full video and the paragraph directly. Meanwhile, we can still utilize the unit-level comparison, *i.e.*, each caption can be used to match the clip first and the video with the most matched clips will be retrieved, *i.e.*, Cap. Avg. (*Caption-Clip*) given a sentence description as a query, we retrieve the video clips directly. For both setups, we use recall as metrics, *i.e.*, R@1, R@5, R@10.

**Dataset.** We evaluate the model pretrained with our TempCLR on YoucookII(Zhou et al., 2018) without any finetuning (*i.e.*, zero-shot). The evaluation set consists of 3350 caption-clip pairs from 436 videos in total. The videos existing in YouCookII have been removed from HT100M.

**Result.** (*Full-Video*) As summarized in Table 1, with Cap. Avg as measurement, when background is removed, VideoCLIP has already outperformed MCN clearly (Exp$_{(3,4)}$). However, as VideoCLIP is not trained to explore the global temporal context, the performance drops when DTW is used as measurement. In contrast, though our approach is only trained with 7.5% of HT100M full set,

Table 4: Performance comparison (%) on action step localization for *Zero-shot* (Left) and *Supervised* (right). TFS: training from scratch[†]: Finetune from VideoCLIP with 7.5% HowTo100M train set.

| Approach (*Zero-shot*) | TFS | Recall | | Approach (*Supervised*) | Recall |
|---|---|---|---|---|---|
| HT100M (Miech et al., 2019) | ✓ | 33.6 | | Alayrac (Alayrac et al., 2016) | 13.3 |
| MIL-NCE (Miech et al., 2020) | ✓ | 40.5 | | Zhukov (Zhukov et al., 2019) | 22.4 |
| MCN (Chen et al., 2021) | ✓ | 35.1 | | Supervised (Zhukov et al., 2019) | 31.6 |
| DWSA (Shen et al., 2021) | ✓ | 35.3 | | VideoCLIP (Xu et al., 2021) | 47.3 |
| UniVL (Luo et al., 2020) | ✓ | 42.0 | | TempCLR (Ours) | **52.5** (↑ **5.2**) |
| VT-TWINS (Ko et al., 2022) | ✓ | 40.7 | | Approach (Few-shot) | Recall |
| VideoCLIP (Xu et al., 2021) | ✓ | 33.9 | | VideoCLIP w/ 20% | 41.1 |
| VideoCLIP[†] | | 33.5 (↓ 0.4) | | TempCLR (Ours) w/ 20% | **42.8** |
| TempCLR (Ours)[†] | | **36.9** (↑ **3.0**) | | | |

benefiting from temporal modelling, TempCLR can effectively improve the performance ($\text{Exp}_{(5,7)}$) without hurting the retrieval between clips and captions ($\text{Exp}_{(4,6)}$). Then, we assume no temporal annotation is provided for full-video retrieval and retrieve full video containing background. As the spatial information in background may distract the sentence features, comparing with the scenario when background is removed, the recall by TempCLR drops. However, from Table 1 and 2, by using either OTAM or DTW for video-paragraph comparison, our TempCLR can outperform the Video-CLIP baseline consistently. (*Caption-Clip*) As summarized in Table 3, by minimizing $\mathcal{L}_{seq}$, the attention mechanism in Transformer is trained to embed global temporal information into each clip feature, which may then facilitate the such unit-level retrieval and achieves slight gain over strong baseline VideoCLIP. For MMFT, the extra audio information can be used to benefit the retrieval.

### 4.2.2 ACTION STEP LOCALIZATION

**Setup and Metric.** Each video is associated with a Task consisting of multiple steps (*i.e.*, sentences). Then, each video frame should be assigned with the corresponding step and we use recall as metric.

**Dataset.** We perform evaluation on CrossTask (Zhukov et al., 2019) and the test set contains 1690 annotated videos over 18 Tasks. We first apply our model pre-trained on HT100M on CrossTask test set for zero-shot evaluation. Then, following Xu et al. (2021), we finetune the VideoCLIP model on 540 videos with our $(L)_{seq}$ and then evaluate the finetuned model on test set (*Supervised*).

**Result.** (*Zero-shot*) As shown in Table 4, when we update VideoCLIP with the 7.5% subset using its original loss ($\text{Exp}_{(8)}$), the model overfits and the performance drops slightly. However, for TempCLR, by adding loss $\mathcal{L}_{seq}$, we can still improve the recall from 33.9 to 36.9. (*Supervised*) VideoCLIP has shown strong performance, but TempCLR can still increase the recall to 52.5. Furthermore, by finetuning on only 100 videos (20% of train set), our approach can effectively improve the performance, which also demonstrates the benefit from exploring the temporal order modelling.

### 4.3 FEW-SHOT ACTION RECOGNITION

**Setup and Metric.** Given a Task where each class has only $N_s$ labeled video as reference (*i.e.*, $N_s$-shot), we classify a test video by comparing its average distance with labeled videos of each class through nearest neighbor search. Following the protocol (Zhu & Yang, 2018), we first pre-train the model on a dataset of classes $\mathcal{C}_b$ and directly evaluated on few-shot tasks which are sampled from another dataset of classes $\mathcal{C}_n$ and $\mathcal{C}_b \cap \mathcal{C}_n = \varnothing$.

Table 5: Performance (%) on action recognition.

| Approach | 1-shot | 5-shot |
|---|---|---|
| TSN++* | 33.6 | 43 |
| CMN++* | 34.4 | 43.8 |
| RTRN++* | 38.6 | 48.9 |
| OTAM (Cao et al., 2020) | 42.8 | 52.3 |
| TRX (Perrett et al., 2021) | 42.0 | 64.6 |
| MTFAN (Wu et al., 2022) | 45.4 | 60.4 |
| TempCLR (Ours) | **47.8** | **67.7** |

*:Results are reported in Cao et al. (2020)

**Dataset**. We use sth-sth V2 (Goyal et al., 2017) for experiment and follow the subset split in Cao et al. (2020). The subset contains 100 classes where $|\mathcal{C}_b| = 64$ and $|\mathcal{C}_n|$ is 24 (12) classes are for evaluation (validation). During evaluation, each Task contains 5 classes and has 15 test videos per class while $N_s = \{1, 5\}$. Finally, we report the mean accuracy over 10k Tasks. As only using spatial content is unreliable, temporal modelling is thus specifically required for correct prediction.

**Results** For fair comparison, we first use a ResNet50 model pretrained on ImageNet (Deng et al., 2009) to extract embedding for each frame which are fed into a Transformer to obtain features in $\mathbf{S}_a$. As for $\mathbf{S}_p$, we set a linear layer to process each frame embedding. Then, we use the pre-trained model as initialization and follow the training of OTAM and TRX. More details can be found in Sec. A.3. TempCLR differs from Cao et al. (2020) by applying self-supervision and using a shuffled version of $\mathbf{S}_p$ as $\mathbf{S}_n$, while they apply meta-training (Snell et al., 2017) between video instances. TRX also employs a Transformer but only learns the temporal order between a pair/triple of frames and the learning efficiency is limited. In contrast, TempCLR directly models the temporal context.

## 5 DISCUSSION

### 5.1 UNIT MATCHING IN SEQUENCE ALIGNMENT

When we directly measure the global distance between video and paragraph, it is also very important to ensure the matched caption-clip pairs are semantically close to each other. As visualized in Fig. 3, for the full video directly retrieved by a paragraph, our approach can also correctly match all captions with the video clips. However, if we only rely on the unit-level similarity, the high visual similarity can cause false prediction. More visualization can be found in appendix.

In addition, as shown in Table 6(a), we check percentage of correctly matched clip-caption pairs averaged over all videos. Then, our TempCLR can correctly match more caption-clip pairs than VideoCLIP when the video is compared with paragraph using DTW. In this way, given a paragraph, our approach can also mitigate the distraction from unpaired videos.

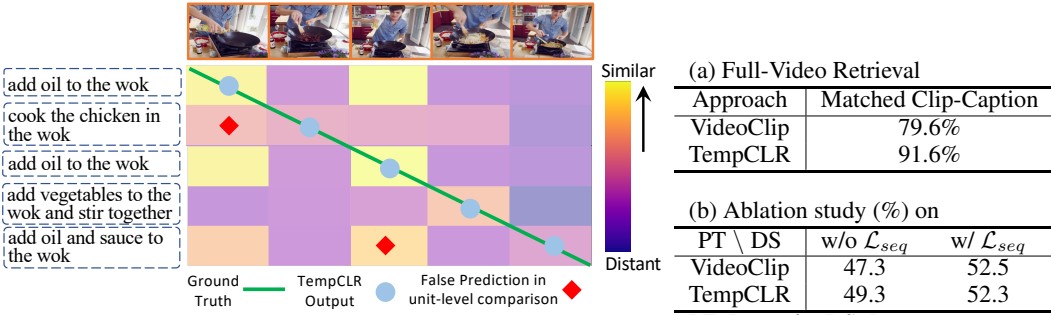

(a) Full-Video Retrieval

| Approach | Matched Clip-Caption |
|---|---|
| VideoClip | 79.6% |
| TempCLR | 91.6% |

(b) Ablation study (%) on

| PT \ DS | w/o $\mathcal{L}_{seq}$ | w/ $\mathcal{L}_{seq}$ |
|---|---|---|
| VideoClip | 47.3 | 52.5 |
| TempCLR | 49.3 | 52.3 |

PT: Pre-train, DS: Downstream

Figure 3: Visualization of video retrieval. TempCLR correctly match all clip-caption pairs while retrieving the full video.

Table 6: Ablation study.

### 5.2 NEGATIVE SAMPLE SELECTION

For each anchor sequence $\mathbf{S}_a$, we generate negative samples $\mathbf{S}_n$ from the positive sample $\mathbf{S}_p$ by 1) considering the difference regarding temporal granularity and 2) shuffling the order to break the temporal consistency. Then, we discuss alternative strategies.

**Video-Paragraph.** For each paragraph (anchor), in addition to shuffle the segments first and then shuffle the clip embeddings within each segment (*seg-unit*), we can also only shuffling the segments while maintaining the clip order within each segment (*seg-only*). Meanwhile, an intuitive strategy is to use the video unpaired with the anchor to build negative samples (*unpaired*) or combine it with *seg-unit joint*. In addition, we can aggressively shuffle all clips embeddings in $\mathbf{S}_p$ (*all-unit*), or only shuffle the clip embeddings within each segment (*within-seg*).

As shown in Table 7, we compare the performance on CrossTask under *Supervised*. (1) Since we use VideoCLIP as initialization and the model has been trained to distinguish clips from different video instances, the sequence-level distance of negative samples in *unpaired* has already been high. Furthermore, as the temporal consistency hardly exists between video and unpaired paragraph, the gain by *unpaired* is inevitably limited. (2) According to $\text{Exp}_{(2-4)}$, since each caption is supposed to align with all of the clips in the paired segment, breaking the order of segments in $\mathbf{S}_p$, *i.e.*, *seg-only* and *seg-unit*, is essential in facilitating model training. From Sec. 5.1, *when two sequences are aligned using DTW, the matched units are also semantically close to each other*. Thus, when

$\mathbf{S}_a$ is compared with $\mathbf{S}_n$, the matched units can indicate the clip features which may cause the most confusion in caption-clip matching due to high visual similarity. In this way, by minimizing $\mathcal{L}_{seq}$, the network is then trained to distinguish those clip features which may hurt the alignment between $\mathbf{S}_p$ and $\mathbf{S}_a$. Then, when the segment order is preserved, comparing with VideoCLIP baseline (*i.e.*, 47.3), *within-seg* results in worse generalization as the confusing clips across segments are not detected and the model can be trained to overfit to trivial difference between clips under the same segment. In contrast, when the segment order is broken, shuffling the clip order within each segment further can serve as data augmentation, which can improve the recall slightly from 52.1 to 52.5. Furthermore, (3) *all-unit* only introduce limited gain since the continuity across clips in one segment is broken and it is too hard for the model to learn. Combining *unpaired* and *seg-unit* in *joint* does not provide clear gain over *seg-unit*. However, we think the main reason is that VideoCLIP has been well-trained for instance discrimination and believe *joint* is still necessary when TempCLR is trained from scratch. Lastly, we can also shuffle the sentence embeddings w.r.t a video (*visual-anchor*) which is equivalent to *seg-only* and achieve reasonably high performance.

Table 7: Ablation study (%) of negative sampling.

| Exp. | Strategy | Recall |
|------|----------|--------|
| 1 | un-paired | 48.0 |
| 2 | within-seg | 46.4 |
| 3 | seg-only | 52.1 |
| 4 | seg-unit | **52.5** |
| 5 | all-unit | 49.3 |
| 6 | joint | **52.5** |
| 7 | visual-anchor | 52.0 |

**Video-Only.** As an alternative, we can also shuffle the frame features of other videos as $\mathbf{S}_n$ and keep training the model in a self-supervised manner. However, since the distance between different video instances has already been high, the model is not well learned and the accuracy is 38.2.

## 5.3 COMPONENT ANALYSIS AND ABLATION STUDY

**Modelling of global temporal context** has been studied for long-video understanding. A popular way is to employ Transformer architecture to model the correlation between clips automatically. However, by explicitly modelling the temporal orders, as demonstrated in Sec. 4, TempCLR is capable to provide consistent gain over three tasks under six different setups from the strong baseline. Thus, the comparison with VideoCLIP already serves as ablation studies to demonstrate the importance of explicit regularization for temporal modelling. After all, the Attention mechanism may require many data to fully understanding the temporal correlation. In addition, specifically for videos, the labels are noisy such as misalignment between ASR transcription and long video (Miech et al., 2020), the attention modelling can also be distracted by background which hurts the training efficiency. As such, our approach provides a general framework aiming to utilize temporal context.

***Supervised* on CrossTask**. As summarized in Table 6, we use CrossTask study the effect of $\mathcal{L}_{seq}$ in pre-training (PT) stage and downstream finetuning (DS) stage. For TempCLR, *i.e.*, with $\mathcal{L}_{seq}$ in PT, as the model has been trained to model global temporal order, finetuning without $\mathcal{L}_{seq}$ can also improve the recall. Meanwhile, though the temporal pattern learned PT may not exactly the same of data in DS, as finetuning with $\mathcal{L}_{seq}$ in DS is very important for down-stream evaluation, the performance are comparable when either VideoCLIP or TempCLR is used for initialization.

**DTW Vs. OTAM**. From Table. 2, as the temporal annotation is given in video-paragraph pre-training, using DTW or OTAM achieves similar performance. However, even when annotation is not given in Video-only task, employing either OTAM or DTW does not impact the performance significantly (47.7 for 1-shot). More details can be found in the appendix.

## 6 CONCLUSION

In this paper, we have proposed TempCLR, a contrastive learning framework to align the temporal dynamics exhibited in video-paragraph pre-training, where the paragraph/video can be represented as a sequence of sentences/clips and are learned to match their global content following the consistent temporal order. Specifically, to encourage temporal modeling over the full sequences and the effectiveness of explicit sequence comparison, we propose to generate negative samples by shuffling clips of the paired video from different temporal granularities. In this way, we can adjustably maximize the agreement between sequences with temporal order consistency and maximize the distance between unaligned sequences with different temporal coherency. Our TempCLR achieves consistent performance gain on three tasks under six different setups, which experimentally demonstrated the effectiveness of our framework. We also provide analysis to validate our design choice of negative sampling, which is shown to both benefits the sequence-level alignment and the unit-level matching.

**Acknowledgement** This material is based on research sponsored by Air Force Research Laboratory (AFRL) under agreement number FA8750-19-1-1000. The U.S. Government is authorized to reproduce and distribute reprints for Government purposes notwithstanding any copyright notation therein. The views and conclusions contained herein are those of the authors and should not be interpreted as necessarily representing the official policies or endorsements, either expressed or implied, of Air Force Laboratory, DARPA or the U.S. Government.

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

# A  APPENDIX

## CONTENTS

## A.1 TERMINOLOGY EXPLANATION

In this paper, we study the video-text pre-pretraining for long-video understanding, and specifically focus on the video-paragraph pre-training. As such, we interchangeably use video-text pre-training and video-paragraph pre-pretraining. Here, the paragraph is essentially a set of captions and each caption is used to describe one part, *i.e.*, segment, in the full video. As such, according the order of video segments, the captions can be organized using the same order and represented as a form of paragraph. As the form of caption is a short sentence, we interchangeably use caption and sentence, such as sentence-caption pair and caption-clip pair.

For pre-training on HT100M (Miech et al., 2019), due to the limitation of computational resources, we essentially finetune from the pre-trained VideoCLIP model using 90k videos (7.5%) of HT100M. However, for the convenience description, we still use pre-training for the training on HW100M and use "finetuning" as fine tuning on the train set of downstream task, *i.e.*, CrossTask (*Supervised*).

In downstream evaluation such as action step localization and few-shot action recognition, we use "Task" to indicate the a localization task (matching each caption in a to different parts in the video) and one few-shot task (classify test videos among the few-shot classes of interest), which is different from "task" used to indicate each of the three task types.

## A.2 IMPLEMENTATION DETAILS FOR TRAINING

### A.2.1 IMPLEMENTATION DETAILS FOR HW100M PRE-TRAINING

**Architecture.** VideoCLIP (Xu et al., 2021) consists of two Transformer architectures for video and paragraph separately. For videos, a backbone network is used to extract embedding for each clip, *i.e.*, video token, where the clip embeddings are used as the input of the six-layer video transformer. For each sentence, VideoCLIP first tokenize the sentences and obtain the token embedding (*e.g.*, the embedding for each word) from embedding lookup (Devlin et al., 2018). For implementation, VideoCLIP use S3D feature (Xie et al., 2018) as the clip embeddings as the S3D network has been pre-trained on HowTo100M (Miech et al., 2019) in a self-supervised manner (Miech et al., 2020). As the embedding dimensions for clips (512) and language tokens (768) are different, an MLP is set preprocess the clip embeddings and increase their dimension. As such, the trainable network components of VideoCLIP includes two Transformer and one MLP layer. During our pre-training, we update the all parameters.

**Dataloading.** As the annotation of temporal segments for long videos is noisy and ASR transcription can also be misaligned with long video (Miech et al., 2020), for each caption, VideoCLIP has designed a strategy

- Extend the timestamps for the caption of interest and the period after extension may include other captions. Then, the newly included captions is concatenated with the caption of interest which are fed in the 12-layer text Transformer.
- Average the Transformer outputs for all of the tokens to obtain one representation for the caption.
- Use the new timestamps to select clip and feed them into 6-layer video Transformer.
- Fuse the information within the segment by averaging the output of video Transformer, *i.e.*, clip features, and obtain a segment representation.

For our approach TempCLR, we reuse its data loading pipeline here to mitigate the potential issue from the noisy annotation.

**Training implementation.** As the features of clips in the same segment has been averaged to obtain one segment representation, during pre-training of our TempCLR on HW100M, we directly shuffle the segment features in $\mathbf{S}_a$ without shuffling the clips features as the segment feature is independent of the clip feature orders. Meanwhile, during the pre-training, we set the batch size of captions is 256 and the captions are sampled from 16 long videos. As we trained from the VideoCLIP pre-trained model, when all parameters are trained, we set the learning rate as $1e^{-5}$. Then, for comparison, as an ablation study, we only update the parameters in the norm layers such as layer norm and the learning rate is set as $2e^{-5}$. In both case, we train the model on 2 NVIDIA

TiTAN RTX GPUs (each with 24 GB memory) for 10 epoches within two hours and use the default hyper-parameters in Adam (Kingma & Ba, 2014) optimizer with betas of (0.9, 0.98).

### A.2.2 IMPLEMENTATION DETAILS FOR CROSSTASK FINETUNING

**Architecture.**     Refer to VideoCLIP architecture.

**Dataloading.**     When the model is finetuned on CrossTask for action step localization, we still keep the embedding of each clip and do not fuse them. As such, we can apply the shuffling strategy described in Fig. 2 and the ablation study is provided in Sec. 5.2.

- For each video, we sample clips by sliding window with window size 16 and each clip takes 32 frames, and load all clips of the video into a batch
- Put all the action steps for the given task into the text encoder to get action step representations. Put the sampled clips to the video encoder and expand the clips to full video in frame level (by returning hidden state).
- Filter out background frames and only keep frames with action step labels.
- Generate negative samples by swapping the action step representations and get the alignment loss.

**Training implementation.**     The learning rate is set as $5e^{-5}$. We use the default hyper-parameters in Adam (Kingma & Ba, 2014) optimizer with betas of (0.9, 0.98)

### A.2.3 IMPLEMENTATION DETAILS FOR VIDEO PRE-TRAINING

**Architecture.**     We use a ResNet50 pretrained on ImageNet to extract embedding for each frame. The Transformer architecture has 6 layers and output the feature for each frame embedding. The linear layer that is set to process each frame embedding to obtain $\mathbf{S}_p$ is also jointly learned.

**Data Loading.**     Then, we follow Cao et al. (2020) to first determine the starting and ending frame index as well as the sampling rate, then we use the selected frame embeddings as the sequence representing the video. According to the starting and ending frame index as well as the sampling rate, we can determine the frames to be sampled.

**Training implementation.**     During training, we only train the Transformer while the ResNet50 is not trained. However, for the methods compared in the Table. 5, the ResNet50 are tuned. The Transformer is trained from scratch and it uses adam optimizer with default hyper-parameter setting. We use cosine annealing schedule and the initial learning rate is 0.001. Meanwhile, we use the weighted sum (0.3, 0.7) of VideoCLIP loss and our $\mathcal{L}_{seq}$ as the final training objective. Since we update the network from the well-trained VideoCLIP parameters, maintaining the original loss of VideoCLIP is to keep the stability of network training. The weight here is also an empirically value that can provide reasonable performance gain and we fix it during the pre-training.

### A.3 DOWNSTREAM TASK EXPLANATION

### A.3.1 VIDEO RETRIEVAL

Video retrieval has been widely studied and its most popular setting is go retrieve video clips given the sentence/caption as a query, (*i.e.*, caption-clip). Recently, to formulate a more realistic scenario, Chen et al. (2021) formulated full-video retrieval, *i.e.*, given a set of caption queries, *i.e.*, paragraph, describing multiple parts of an entire long video, the task then aims to retrieve the full video according to the paragraph. However, for each long video, Chen et al. (2021) directly concatenate all video clips which has paired sentences. Then, a strong assumption is applied that the background can be removed according to the temporal annotation. As such, we take a step further to consider the full-video retrieval without any temporal annotation and the background are included. The performance is measured by recall, the number of captions/paragraphs that can correctly retrieve the clips/videos. For each language query, we will rank the retrieving result according to the similarity and a correct assignment means the video is correctly retrieved among the videos with top k high similarity, *i.e.*, R@K, and we have $k = \{1, 5, 10\}$

### A.3.2 ACTION STEP LOCALIZATION

This task assumes that each video is associated with a T̲ask consisting of multiple steps. During inference, given a T̲ask and its corresponding step candidates, each video frame/clip is supposed to be assigned to the corresponding step. This assignment is determined according to the similarity between the frame/clip feature with the caption feature, *i.e.*, in a form of classification task. Then, we directly use recall R@1, *i.e.*, number of step assignments that fall into the correct ground truth interval, divided by the total number of steps in the video, as the metric.

### A.3.3 FEW-SHOT ACTION RECOGNITION

For few-shot learning, we are first given a *base* dataset of classes $\mathcal{C}_{base}$ and a *novel* dataset of classes $\mathcal{C}_{novel}$ where the two class sets are *disjoint*, *i.e.*, $\mathcal{C}_{base} \cap \mathcal{C}_{novel} = \emptyset$. The *base* set is used to pre-train a model and the *novel* dataset is used to sample few-shot T̲asks for evaluation.

In each T̲ask, there are $N_c$ classes and each class has $N_s$ training samples, *i.e.*, $N_c$-way $N_s$-shot, and the training samples are termed as support, which is used to differentiate from the training data in fully-supervised learning. Also, the support may not be used to finetune the network. In prototype classification, the features of support videos are averaged for each class and the averaged features are used as prototype, *i.e.*, the weights in the classifier, and the classification is done through nearest neighbor search. Then, the test data within each T̲ask is termed as query. Usually, during evaluation, each class have 15 queries to be classified, *i.e.*, $15N_C$ queries in total. During testing, for our approach, since we cannot aggregate the sentences into one sentence, we just calculate the distance between the query and all support videos, and use the distance averaged for each class is reference for classification.

Following the common setting (Zhu & Yang, 2018), we set $N_c = 5$ and $N_s = \{1, 5\}$, *i.e.*, 5-way 1-shot and 5-way 5-shot. We then report the mean accuracy over 10,000 tasks for each task type. To note, as $\mathcal{C}_{base} \cap \mathcal{C}_{novel} = \emptyset$, and no finetuning on support videos in *novel* is applied, the recognition accuracy can be used to indicate the generalization ability of our approach on new classes.

For the dataset something-something v2 used in our experiment, and class typically asks the relationship between objects, such as something pushes something and the objects can vary from each other in different video instanes. As such, successful recognition on this dataset specifically requires the understanding of global temporal information.

Table 8: Detailed results of performance (%) on action recognition

| Accuracy | 1-shot | 5-shot |
|---|---|---|
| TempCLR w/ OTAM | 47.8 | 62.9 |
| TempCLR w/ TRX | 45.1 | 67.7 |

For the results reported in few-shot action recognition, we use the model pre-trained under our self-supervision pre-training objective as the initialization and then use the loss proposed in OTAM or TRX for meta-training. The details of the results are reported in Table 8. We note that the initialization for all experiments and all approaches are the same for fair comparison. In this way, we highlight that our approach is orthogonal to the existing meta-training approaches and can significantly improve the few-shot adaptation efficiency. Meanwhile, the performance of TempCLR w/ OTAM on 5-shot is limited by the computational resources by the time of submission. We will provide more results when we access more GPU resources.

### A.4 MORE EXPERIMENTAL RESULTS

### A.4.1 MEASUREMENT

(*Cap. Avg.*) still utilizes the unit-level comparison, *i.e.*, each caption can be used to match the clip first. Then, for each video, the number of clips video retrieved by the captions in the set is used to indicate the similarity between the paragraph and the full video

(*DTW*) Please refer to Sec. 3.2. In particular, for the matching between units, DTW has a strong assumption that $M(1, 1) = M(N_a, N_p) = 1$ is always met.

(*OTAM*) stands for Ordered Temporal Alignment Module. To avoid the boundary assumption in DTW, Cao et al. (2020) proposed OTAM. The basic idea is to pad the sequence with meaningless unit such as zero vector at the begining and end part. In this way, when dynamic programming is resued to calculate the matching, the first (last) units in the two sequences to be compared are not necessary matched. For detail about the implementation of OTAM, please refer to Cao et al. (2020).

### A.4.2 VIDEO RETRIEVAL ON DIDEMO

We conduct video retrieval experiments on another dataset DiDeMo (Anne Hendricks et al., 2017), which contains 10,000 videos annotated with 40,000 sentences on Flicker videos. We follow Video-CLIP (Xu et al., 2021) and conduct evaluation on 4021 test samples.

As summarized in Table 9, we first conduct experiments on the clip-caption retrieval setting. We follow VideoCLIP and summarized the performance in both zero-shot and supervised. Our Temp-CLR can consistently improve the performance than the VideoCLIP baseline and even outperform most supervised-based approaches. In particular, the performance at R@5 also outperform the best performance of ClipBERT.

Table 9: Performance (%) comparison on video retrieval on DiDeMo (clip-caption)

| Approach | R@1 | R@5 |
|---|---|---|
| Supervised | | |
| S2VT (Venugopalan et al., 2014) | 11.9 | 33.6 |
| FSE (Zhang et al., 2018) | 13.9 | 44.5 |
| CE (Liu et al., 2019) | 16.1 | 41.1 |
| ClipBERT (Lei et al., 2021) | 20.4 | 48.0 |
| Zero-shot transfer | | |
| VideoCLIP (Xu et al., 2021) | 16.6 | 46.9 |
| VideoCLIP$^\dagger$ | 16.4 | 47.1 |
| TempCLR | **17.7** | **48.1** |

Then, as compared in Table 10 & 11, we conduct experiments on full-video retrieval (background removed). As the domain between HowTo100M can be distant from the domain of DiDeMo, the retrieval scores are generally lower than those on YouCookII, though the video in DiDeMo is shorter. In this case, we observe that exploring the temporal is very useful for the VideoCLIP baseline and our TempCLR. When the background is removed, Then, our approach TempCLR can consistently improve the performance by using DTW or OTAM as measure. In particular, using OTAM as measure can already achieve higher performance than using Cap. Avg. as measure based on unit-level comparison. At the same time, with Cap. Avg as measure, the retrieval performance is also improved comparing with VideoCLIP baseline. Then, we also perform metric ensembling. For both VideoCLIP and TempCLR, employing metric ensembling can help improve the performance while our TempCLR is consistently better than the VideoCLIP baseline. Finally, for full-video retrieval (background kept), for both VideoCLIP and TempCLR, using either DTW or OTAM as measure can improve the performance consistently than using Cap. Avg. as the measure. Again, as mentioned before, the domain gap between HowTo100M and DiDeMo may make the network difficult to generalize well, which then demonstrates the importance of using temporal context and perform sequence-level comparison directly. Then, TempCLR achieves the best performance when the metrics are ensembled.

### A.4.3 FULL-VIDEO RETRIEVAL

As shown in Table 12, we comprehensively study the performance of our approach under different metrics. Again, as VideoCLIP is not trained to explore the global temporal context, there is a large performance drop when DTW or OTAM is used to measure the distance. For video retrieval where the background is removed, the number of features in the paragraph part and the video part are the same where the features with the same index should be matched. As DTW has a strong assumption that $M(1,1) = M(N_a, N_p) = 1$, such prior constraint can benefit the matching.

However, for our approach TempCLR, when the model has been trained to compare the sequences directly, using either OTAM or DTW and always achieve good performance. In addition, Cap. Avg

Table 10: Performance (%) comparison of full-video retrieval on DiDeMo (background removed)

| Measure | R@1 | R@5 | R@10 |
|---|---|---|---|
| VideoCLIP | | | |
| DTW | 8.5 | 24.4 | 37.7 |
| OTAM | 8.9 | 26.6 | 38.6 |
| DTW + Cap. Avg. | 10.3 | **28.2** | 38.6 |
| OTAM + Cap. Avg. | 10.6 | 27.7 | 38.8 |
| Cap. Avg. | 8.9 | 27.0 | 37.9 |
| TempCLR | | | |
| DTW | 9.2 | 26.6 | 39.4 |
| OTAM | 10.4 | 25.6 | 39.9 |
| DTW + Cap. Avg. | 9.9 | 26.3 | 38.4 |
| OTAM + Cap. Avg. | **10.9** | 25.6 | **40.6** |
| Cap. Avg. | 9.4 | 26.3 | 36.4 |

Table 11: Performance (%) comparison of full-video retrieval on DiDeMo (background kept)

| Measure | R@1 | R@5 | R@10 |
|---|---|---|---|
| VideoCLIP | | | |
| DTW | 8.6 | 21.0 | 31.2 |
| OTAM | 8.1 | 22.1 | 31.7 |
| DTW + Cap. Avg. | 9.0 | 22.6 | 32.4 |
| OTAM + Cap. Avg. | 8.3 | 22.6 | 32.6 |
| Cap. Avg. | 7.1 | 21.0 | 31.9 |
| TempCLR | | | |
| DTW | 9.0 | 21.4 | 31.0 |
| OTAM | 9.3 | 21.4 | 31.4 |
| DTW + Cap. Avg. | 9.0 | 21.7 | 31.2 |
| OTAM + Cap. Avg. | **9.5** | **21.7** | **31.4** |
| Cap. Avg. | 8.1 | 21.7 | 32.8 |

is unit-level based metric and can serve as a complementary component for video retrieval. As such, DTW/OTAM and can combined with Cap. Avg and the prediction by the combined metric can also achieve very high recall. Nevertheless, we believe the temporal alignment between two sequences still take the dominant role since the performance by the ensembled metric does not provide significant gain over the recall by using DTW/OTAM only.

Table 12: Ablation study on full-video retrieval (Background Removed)

| Measure | R@1 | R@5 | R@10 |
|---|---|---|---|
| VideoCLIP | | | |
| DTW | 56.0 | 89.9 | 96.3 |
| OTAM | 52.8 | 89.2 | 95.0 |
| DTW + Cap. Avg. | 85.8 | 97.7 | 99.1 |
| Cap. Avg. | 74.5 | 94.1 | 97.9 |
| TempCLR | | | |
| DTW | 83.5 | 97.2 | 99.3 |
| OTAM | 84.9 | 97.9 | 99.3 |
| DTW + Cap. Avg. | 86.5 | 97.2 | 99.3 |
| Cap. Avg. | 74.5 | 94.6 | 97.0 |

Then, the performance of full-video retrieval (background kept) is summarized in Table. 13. As VideoCLIP has been well-trained on the full HowTo100M dataset and compare clip-caption pairs across videos, VideoCLIP can thus discriminate clips of background and can maintain high performance (*i.e.*, 73.6) with Cap. Avg. as the measure. However, as the clips of background accounts for most part of the video and VideoCLIP is not trained to model the temporal context, the perfor-

mance drop clearly when DTW or OTAM is used as measure. Furthermore, when we ensemble the DTW-related measure with Cap. Avg., the performance improvement is thus limited (*i.e.*, from 73.6 to 74.8 by OTAM and 73.8 by DTW).

However, when we train the TempCLR on a very small dataset, the model can then capture the temporal context information effectively. We note that DTW has strong boundary assumption and thus the performance is limited. However, OTAM can avoid the assumption and can then improve R@1 of TempCLR from 70.4 to 72.2. In this way, we believe necessary modification on DTW-related measure can be done to further improve the training efficiency and we leave this for future work. In addition, we can achieve higher performance when the measures are combined. For example, by combining OTAM and Cap. Avg., the performance can be improved to 77.5, which further demonstrates that purely using Cap. Avg. as measure is not enough and it is necessary to explore the temporal context in feature embedding.

Table 13: Ablation study on full-video retrieval (Background Kept)

| Measure | R@1 | R@5 | R@10 |
|---|---|---|---|
| VideoCLIP | | | |
| DTW | 55.7 | 93.1 | 98.9 |
| OTAM | 56.6 | 92.8 | 98.9 |
| DTW + Cap. Avg. | 73.8 | 95.4 | 98.6 |
| OTAM + Cap. Avg. | 74.5 | 95.1 | **99.1** |
| Cap. Avg. | 73.6 | 94.7 | 98.4 |
| TempCLR | | | |
| DTW | 70.4 | 93.8 | 97.9 |
| OTAM | 72.2 | 94.5 | 97.7 |
| DTW + Cap. Avg. | 76.7 | 95.6 | 98.4 |
| OTAM + Cap. Avg. | **77.5** | **95.6** | 98.6 |
| Cap. Avg. | 71.7 | 94.5 | 97.9 |

### A.4.4 ACTION STEP LOCALIZATION

As VideoCLIP employs a Transformer architecture, we also study two different training strategies, *i.e.*, updating all parameters and updating the layer norms only.

(*Zero-Shot*) The results in Table 14 (Left) study the performance by training on HW100M using either OTAM or DTW as to measure sequence distance in TempCLR. It is observed that finetuning all parameters can improve the zero-shot adaptation performance most and using either OTAM or DTW achieves similar performance. However, if only the norm layer is updated, the trianing efficiency by using OTAM as sequence distance measurement can be slightly less effective.

(*Supervised*) For the finetuning on CrossTask, we always update the whole network for fair comparison. However, the pre-trained model by our TempCLR is obtained by updating either all parameters or norm layer only during the pre-traning stage. The two pre-trained models uses DTW as measurement in $\mathcal{L}_{seq}$. In the Table 14 (Right), each column indicates how the finetuning loss is set where w/o $\mathcal{L}_{seq}$ indicates the vanilla contrastive-learning based approach between captions and clips which does not consider the global temporal context.

Then, we can find that 1) using DTW during finetuning is better than using OTAM and 2) updating the layer norm only can stably improve the performance with small variance. When DTW is used and all parameters are finetuned, the model can be well-learned during the pre-training stage and thus achieves good performance in the *Supervised* setup.

Table 14: Ablation study on action step localization under *Zero-Shot* (Left) and *Supervised* (Right).

| (*Zero-Shot*) | OTAM | DTW |
|---|---|---|
| TempCLR + All Params. | 36.9 | 36.9 |
| TempCLR + Norm Only | 36.1 | 36.8 |
| The performance of VideoCLIP is 33.9 | | |

| (*Supervised*) | w/o $\mathcal{L}_{seq}$ | OTAM | DTW |
|---|---|---|---|
| VideoCLIP | 47.3 | 51.7 | 52.5 |
| TempCLR + All Params. | 49.3 | 50 | 52.3 |
| TempCLR + Norm Only | 49.0 | 51.1 | 51.6 |

Then, we provide the full table of *Zero-Shot* in Table 15. As mentioned in the main paper, Video-CLIP (Xu et al., 2021) does not provide clear performance gain over HT100M and under-performs many other SOTA approach in zero-shot evaluation. Since we only have 90k training data and is trained from VideoCLIP, the model by our approach is hard to be well-learned for the zero-shot localization evaluation. However, our approach can still achieve 9% gain and improve the recall from 33.9 to 36.9. Nevertheless, as summarized in Table 16, by finetuning from 50 to 100 training data on CrossTask, we can effectively improve the performance.

Table 15: Full table for Action Step Localization (*zero-shot*)

| Approach (*Zero-Shot*) | Backbone | Recall |
|---|---|---|
| HT100M (Miech et al., 2019) | R152 + RX101 | 33.6 |
| MIL-NCE (Miech et al., 2020) | S3D-G | 40.5 |
| MIL-NCE (Miech et al., 2020) | I3D-G | 36.4 |
| ActBERT (Zhu & Yang, 2020) | R101 + Res3D | 37.1 |
| ActBERT (Zhu & Yang, 2020) | + Faster-RCNN | 41.4 |
| UniVL (Luo et al., 2020) | S3D-G | 42.0 |
| MCN (Chen et al., 2021) | R152 + RX101 | 35.1 |
| MMFT (Shvetsova et al., 2022) | R152 + RX101 | 39.3 |
| MMFT (Shvetsova et al., 2022) | S3D-G | 41.1 |
| VideoCLIP (Xu et al., 2021) | S3D-G | 33.9 |
| TempCLR (Ours) | S3D-G | 36.9 (↑ **3.0**) |

Table 16: Few-shot fine-tuning for action step localization

| Approach | 50(10%) | 100(20%) |
|---|---|---|
| VideoCLIP (Xu et al., 2021) | 40.1 | 41.1 |
| TempCLR (Ours) | 40.2 | 42.8 |

### A.4.5 DISCUSSION ON NEGATIVE SIZE

As summarized in Table 17, we adjust the size of negative samples from 16 to 64 and use the zero-shot performance on action step localization and full-video retrieval for comparison. We use Recall (%) as metric for action step localization and R@1 (%) for full-video retrieval evaluation. By increasing the negative size from 16 to 32, we can see the performance can be improved slightly. However, when we keep increasing the size to 64, there is no significant gain. To note, during pre-training, as we directly use the data sampling strategy in VideoCLIP, the number of segments existing in each video is only 6 on average. To reach reasonably high performance but still train the network efficiently, we set the size of negative samples as 32.

### A.4.6 DISCUSSION ON UNIT MATCHING IN SEQUENCE ALIGNMENT

From the results in Table 12 and Table 6(a), video clips relies mainly replies on the unit-level comparison for video retrieval (hige recall by Cap. Avg.). Even though the percentage of corrected matched caption-clips between the paragraph and video in the pair reaches 80%, the recall is still low when DTW/OTAM is used. This is because the set of sentences features can also have high alignment score with (*i.e.*, distracted by) clip features from other videos, which further demonstrate that the success of VideoCLIP is mainly based on unit-level comparison. However, our approach TempCLR improves both recall and the percentage of correctly matched caption-clip pairs, which indicates that the model has indeed been trained to model the global temporal correlation.

Table 17: Ablation study of Negative Size in Zero-shot transfer learning (%)

| Task \ Negative Size | 16 | 32 | 64 |
|---|---|---|---|
| action step localization (Recall) | 36.5 | 36.9 | 37.1 |
| full-video retrieval (R@1) | 82.9 | 83.5 | 83.5 |

A.5   COMPARISON OF OPTIMIZATION GRADIENT BETWEEN TEMPCLR AND VIDEOCLIP

In this section, to explain the difference of training objective in VideoCLIP and TempCLR, we compare the gradient by obtained by different training strategies, temporally overlapped strategy by VideoCLIP (Xu et al., 2021) and the DTW-based sequence-level comparison by TempCLR. We take a toy example where two sequences $M = [m_1, m_2]$ and $N = [n_1, n_2]$ should be aligned and the ground-truth of matched pairs are $(m_1, n_1)$ and $(m_2, n_2)$. Without loss of generality, we set $M$ as anchor and $N$ as query. Then, the training losses are,

(for VideoCLIP)

$$\mathcal{L}_{VideoCLIP} = \mathcal{L}_1 + \mathcal{L}_2 = -\log \frac{e^{m_1 \cdot n_1^T}}{e^{m_1 \cdot n_1^T} + e^{m_1 \cdot n_2^T}} - \log \frac{e^{m_2 \cdot n_2^T}}{e^{m_2 \cdot n_2^T} + e^{m_2 \cdot n_1^T}}$$

where $\mathcal{L}_1$ and $\mathcal{L}_2$ are the losses of positive pairs for anchor $m_1$ and $m_2$ in $M$ respectively,

and (for TempCLR)

$$\mathcal{L}_{TempCLR} = -\log \frac{e^{m_1 \cdot n_1^T + m_2 \cdot n_2^T}}{e^{m_1 \cdot n_1^T + m_2 \cdot n_2^T} + e^{m_1 \cdot n_2^T + m_2 \cdot n_1^T}}$$

where $\mathcal{L}_{TempCLR}$ is the loss for $M$.

Then, we take the positive pair $(m_1, n_1)$ as one example and derive the gradients w.r.t. $m_1 \cdot n_1^T$,

(For VideoCLIP)

$$\frac{\partial \mathcal{L}_{VideoCLIP}}{\partial (m_1 \cdot n_1^T)} = \frac{\partial \mathcal{L}_1}{\partial (m_1 \cdot n_1^T)} = -e^{m_1 \cdot n_1^T} \frac{e^{m_1 \cdot n_2^T}}{e^{m_1 \cdot n_1^T}(e^{m_1 \cdot n_1^T} + e^{m_1 \cdot n_2^T})} = \frac{-e^{m_1 \cdot n_2^T}}{e^{m_1 \cdot n_1^T} + e^{m_1 \cdot n_2^T}}$$

and (For TempCLR)

$$\frac{\partial \mathcal{L}_{TempCLR}}{\partial (m_1 \cdot n_1^T)} = -e^{m_1 \cdot n_1^T + m_2 \cdot n_2^T} \frac{e^{m_1 \cdot n_2^T + m_2 \cdot n_1^T}}{e^{m_1 \cdot n_1^T + m_2 \cdot n_2^T}(e^{m_1 \cdot n_1^T + m_2 \cdot n_2^T} + e^{m_1 \cdot n_2^T + m_2 \cdot n_1^T})}$$

which can be further simplified as

$$\frac{\partial \mathcal{L}_{TempCLR}}{\partial (m_1 \cdot n_1^T)} = \frac{-e^{m_1 \cdot n_2^T}}{e^{m_1 \cdot n_1^T} e^{m_2 \cdot n_2^T - m_2 \cdot n_1^T} + e^{m_1 \cdot n_2^T}}$$

As such, for our TempCLR, we can observe that the optimization of the positive pair $m_1 \cdot n_1^T$ clearly considers the difference when $m_2$ is matched with each element in $N$, $i.e.$, $e^{m_2 \cdot n_2^T - m_2 \cdot n_1^T}$. In other words, the optimization of $m_1 \cdot n_1^T$ also depends on the correlation between $m_2 \cdot n_2^T$ and $m_2 \cdot n_1^T$, which is the temporal context.

$$\frac{\partial \mathcal{L}_{VideoCLIP}}{\partial (m_1 \cdot n_1^T)} \Big/ \frac{\partial \mathcal{L}_{TempCLR}}{\partial (m_1 \cdot n_1^T)} = \frac{(e^{m_1 \cdot n_1^T} + e^{m_1 \cdot n_2^T})^{-1}}{(e^{m_1 \cdot n_1^T} e^{m_2 \cdot n_2^T - m_2 \cdot n_1^T} + e^{m_1 \cdot n_2^T})^{-1}}$$

We then take $m_1 \cdot n_2^T$ as one example and compare the optimization of negative pairs. (For VideoCLIP)

$$\frac{\partial \mathcal{L}_{VideoCLIP}}{\partial (m_1 \cdot n_2^T)} = \frac{\partial \mathcal{L}_1}{\partial (m_1 \cdot n_2^T)} = \frac{e^{m_1 \cdot n_2^T}}{e^{m_1 \cdot n_1^T} + e^{m_1 \cdot n_2^T}}$$

and (For TempCLR)

$$\frac{\partial \mathcal{L}_{TempCLR}}{\partial (m_1 \cdot n_2^T)} = \frac{e^{m_1 \cdot n_2^T + m_2 \cdot n_1^T}}{e^{m_1 \cdot n_1^T + m_2 \cdot n_2^T} + e^{m_1 \cdot n_2^T + m_2 \cdot n_1^T}}$$

Similarly, we can also observe that optimization of negative pairs only considers unit-level comparison $m_1 \cdot n_1^T$ and $m_2 \cdot n_2^T$ in VideoCLIP while ours also takes temporal context, $i.e.$, the matching difference $m_2 \cdot n_2^T - m_2 \cdot n_1^T$ into consideration.

$$\frac{\partial \mathcal{L}_{VideoCLIP}}{\partial (m_1 \cdot n_2^T)} \Big/ \frac{\partial \mathcal{L}_{TempCLR}}{\partial (m_1 \cdot n_2^T)} = \frac{(e^{m_1 \cdot n_1^T - m_1 \cdot n_2^T} + 1)^{-1}}{(e^{m_1 \cdot n_1^T - m_1 \cdot n_2^T + m_2 \cdot n_2^T - m_2 \cdot n_1^T} + 1)^{-1}}$$

### A.6 MORE VISUALIZATION & FIGURE EXPLANATION

We first provide the full version of visualization shown in Fig. 3 in Fig. 4. Then, we provide three more visualization results in Fig. 5 & 6 & 7. The visualization demonstrates that our approach is capable to correctly match the clip-caption pairs with high confidence and also achieve sequence alignment for full-video retrieval.

Meanwhile, in our Concept Figure 1, as a comparison between sequence-level distance and unit-level distance, (b) has similar color pattern to (a) since the color of chicken is dark, which results in difficult and potential mismatching under unit-level comparison.

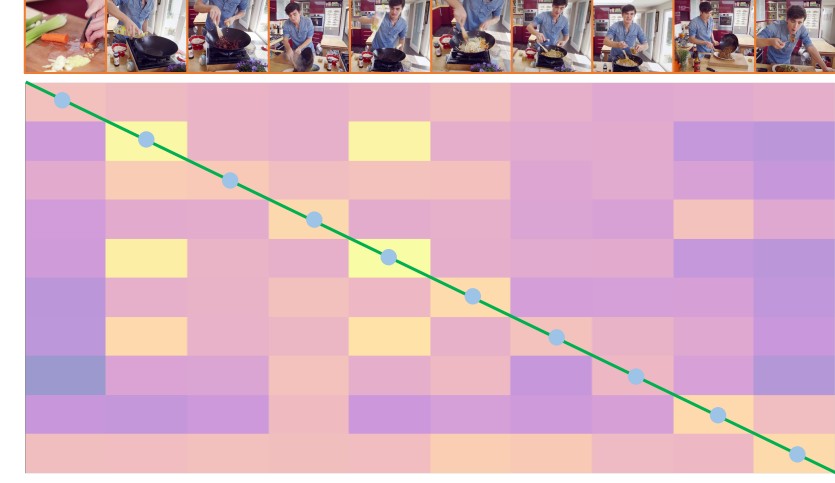

1. chop vegetables

2. add oil to the wok

3. cook the chicken in the wok

4. transfer the chicken to a plate

5. add oil to the wok

6. add vegetables to the wok and stir

7. add oil and sauce to the wok

8. add chicken noodles and seasoning to the wok and stir together

9. transfer everything onto a platter

10. Anno taste the stir fry

Figure 4: Full version of visualization in the main paper.

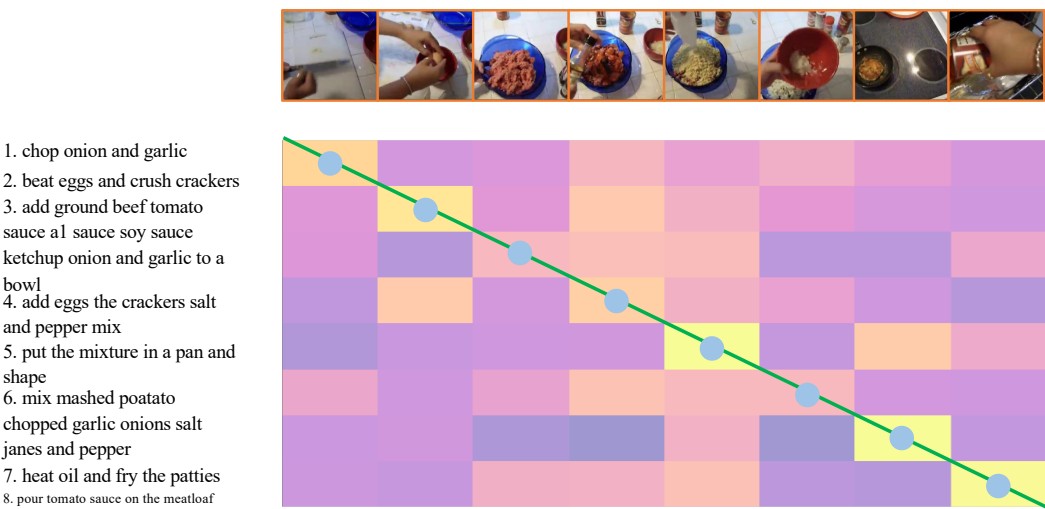

1. chop onion and garlic

2. beat eggs and crush crackers

3. add ground beef tomato sauce a1 sauce soy sauce ketchup onion and garlic to a bowl

4. add eggs the crackers salt and pepper mix

5. put the mixture in a pan and shape

6. mix mashed poatato chopped garlic onions salt janes and pepper

7. heat oil and fry the patties

8. pour tomato sauce on the meatloaf

Figure 5: Additional visualizaton examples.

Then, we examine the robustness of our TempCLR towards scene change. As shown in Fig. 8, we consider the case where the change of scene/context does not break the temporal succession. Given a full video which consists of 10 captions and each caption is paired with a segment in the full video. Then, we replace the video segment corresponding to the second step "add oil to the wok" as a segment which is also describing "add oil to the wok" but from another video. As we feed the whole video into the network jointly, the interaction between the clips of the replaced segment with others may adjust the extracted features, comparing with Fig. 4, the similarity between the captions

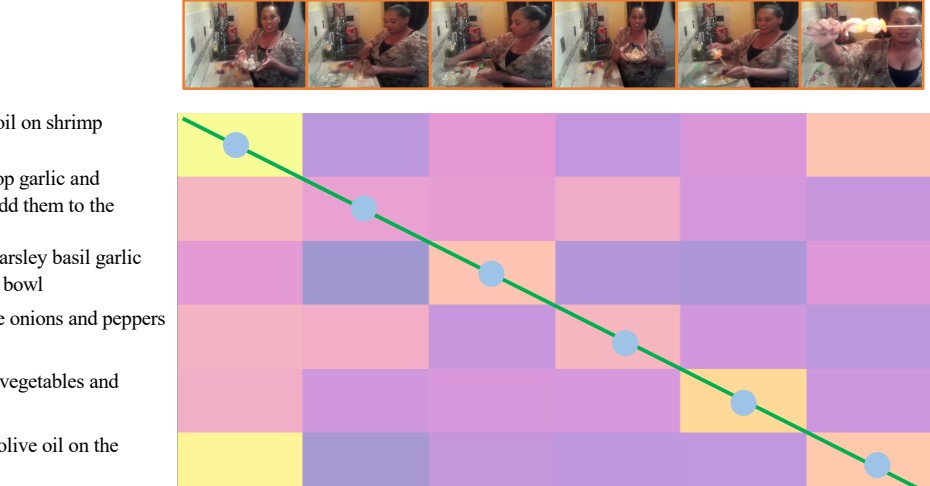

1. chop onions equivalent to the quantity of beef meat

2. heat some oil in a deep pan and add chopped onions and fry till they turn brown
3. clean and chop 1 whole garlic
4. stir and let onions fry till they are completely brown

5. once the onions are done add garlic and stir
6. add the diced beef meat to it and roast it
7. add pepper cumin salt paprika red pepper powder stir everything and roast
8. add a cup of tomato puree and cook stirring
9. pour some warm water to it and cook until the meat becomes soft
10. melt some better in another small pan and add 3 tsp of flour to it and mix and roast until it browns
11. add the roasted flour to the cooking meat mixture and mix well and cook for 10 minutes more
12. finely chop marjoram and add it to the goulash
13. plate some bread pour the goulash garnish with some chopped onions and serve

Figure 6: Additional visualizaton examples.

1. pour olive oil on shrimp

2. roughly chop garlic and peppers and add them to the bowl
3. add some parsley basil garlic powder to the bowl

4. cut up some onions and peppers into squares

5. skewer the vegetables and shrimps

6. pour some olive oil on the kabob

Figure 7: Additional visualizaton examples.

and the original video clips are changed slightly. However, with DTW as measure, as the steps in the videos are still successive, we can still align the captions with videos correctly.

Next, we examine the robustness towards background (*i.e.*, clips without clear semantic meaning), as shown in Fig. 9, as inject a segment of background between the 4th and 5th steps in the original video. Then, as the features of all captions are distant from the background, it does not impact the optimal alignment derived in DTW. As such, our TempCLR is still robust to the background.

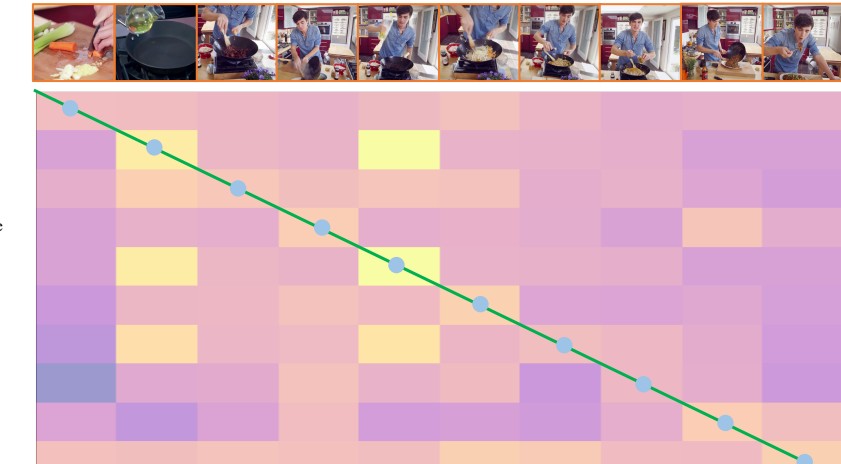

1. chop vegetables

2. add oil to the wok

3. cook the chicken in the wok

4. transfer the chicken to a plate

5. add oil to the wok

6. add vegetables to the wok and stir

7. add oil and sauce to the wok

8. add chicken noodles and seasoning to the wok and stir together

9. transfer everything onto a platter

10. taste the stir fry

Figure 8: Visualization on scene change.

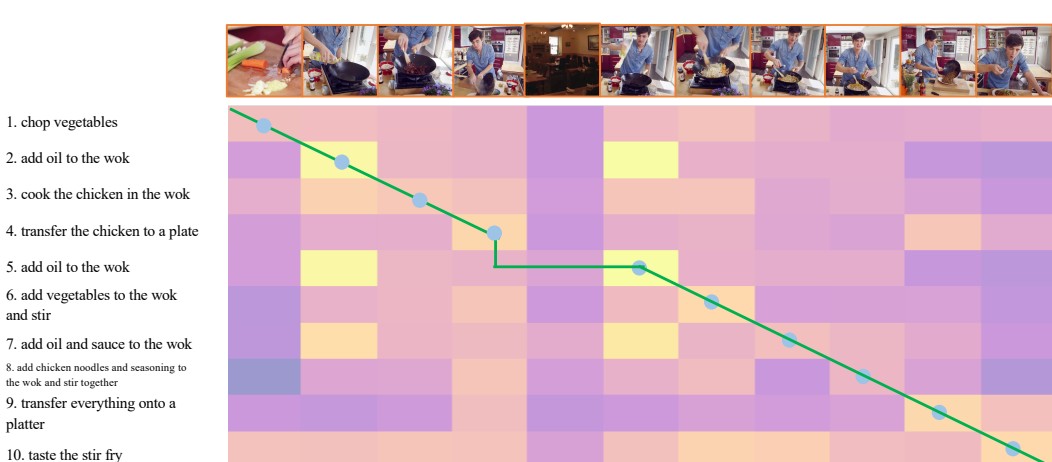

1. chop vegetables

2. add oil to the wok

3. cook the chicken in the wok

4. transfer the chicken to a plate

5. add oil to the wok

6. add vegetables to the wok and stir

7. add oil and sauce to the wok

8. add chicken noodles and seasoning to the wok and stir together

9. transfer everything onto a platter

10. taste the stir fry

Figure 9: Visualization on background change.

