# OpenReview forum: "TempCLR: Temporal Alignment Representation with Contrastive Learning"
_ICLR.cc/2023/Conference — ICLR 2023 poster_

### Official Review · Reviewer_e5YM · 2022-10-20

**Confidence:** 4
**Correctness:** 3
**Technical Novelty And Significance:** 2
**Empirical Novelty And Significance:** 3
**Recommendation:** 6

**Clarity, Quality, Novelty And Reproducibility:**

Clarity: The paper writing is hard to understand. For example, in table 4, '$^*$' of VideoCLIP$^*$ misses notation. The grammar and the usage of the vocabulary also need to be improved.

Reproducibility: The implementation details are listed in the supplementary appendix thoroughly.



**Strength And Weaknesses:**

Strength:
+ The empirical results achieve SOTA performance across three tasks under six different setups.
+ The methodology is straightforward and makes sense. Unit-level alignment is important for downstream tasks, e.g., text retrieval.

Weakness:
+ Compared to VideoCLIP[1], the technique contribution seems weak. I wonder about the actual effect of DTW compared to the temporally overlapped pairs strategy. Besides, the random shuffling trick is not novel in self-supervised video representation learning [2].
+ The writing is hard to follow due to typos and needs to be polished. For example, the caption of Table 2. 'Mesure' --> 'Measure'.


[1] VideoCLIP: Contrastive Pre-training for Zero-shot Video-Text Understanding. EMNLP 2021.

[2] Self-supervised Spatiotemporal Learning via Video Clip Order Prediction. CVPR 2019.

**Summary Of The Paper:**

The proposes a novel contrastive learning pipeline for text-video correspondence. The method adopts dynamic time warping to align the sequence-level consistency and takes shuffled clips as the negative pairs. The experiments on video retrieval, action step localization, and few-shot action recognition have shown the method achieves effective representation. In addition, the ablation study also dissects the core strategies in the paper like DTW and negative sample selection.

**Summary Of The Review:**

Overall, the contribution of the paper is incremental and lacks novelty. Thus, I recommend rejecting the paper.

---

> ### Author Response · Authors · 2022-11-19
> **Response to Reviewer e5YM (1/2)**
>
> **Question 1**: Comparison with strategy based on temporally overlapped pairs in VideoCLIP.
>
> A1: We summarize and explain the effect of DTW in TempCLR comparing to temporally overlapped pairs in VideoCLIP from the following aspects.
>
> (*Sequence-level Vs. Unit-level*) *VideoCLIP* compares the similarity between clips and captions individually. For example, given the video and its paired paragraph, VideoCLIP is trained to maximize the similarity between all matched caption-clip pairs and aggressively minimize the similarities with other clips from the same video. In this way, the optimization of each pair is independent from the content of the video/paragraph.
> In contrast, for *TempCLR*, the optimization of each matched caption-clip pair depends on the content of entire video/paragraph (comparison of gradient is provided below). Under DTW, the determination of matching in one clip-caption pair also depends on the distance of other clip-caption pairs. Thus, we expect the matched caption-clip pairs derived from DTW should also be temporally overlapped (detailed in experiment below). In this way, the model is trained to extract features which are aware of the temporal context. Though we found it is not necessary to apply this constraint during training, we think this is benefited from the VideoCLIP initialization and contrastive learning between videos and we think it is still necessary when the model is trained from scratch.
>
> (*Experimental Study*) Experiments on three tasks have clearly shown the benefit of employing DTW to compare sequence-level distance during trianing.
> By using a small set of HowTo100M to train from VideoCLIP, we can achieves consistent performance gain.
> Furthermore, as shown in Table 6(a), by embedding the temporal context into features, in video retrieval task, when we compare the video and paired paragraph, we can also successfully match the clip-caption pairs.
>
> (*Optimization*) We compare the gradient by two different training objectives and explain the effect of DTW in our TempCLR. We take a toy example where two sequences $M=[m_1,m_2]$ and $N=[n_1,n_2]$ should be aligned and the ground-truth of matched pairs are $(m_1,n_1)$ and $(m_2,n_2)$. Without loss of generality, we set $M$ as anchor and $N$ as query. Then, the training objectives/loss are,
>
> (for VideoCLIP)
>
> $$
> \mathcal{L}_{VideoCLIP} = \mathcal{L}_1 + \mathcal{L}_2 = -\log\frac{e^{m_1 \cdot n_1^T}}{e^{m_1 \cdot n_1^T} + e^{m_1 \cdot n_2^T}} - \log\frac{e^{m_2 \cdot n_2^T}}{e^{m_2 \cdot n_2^T} + e^{m_2 \cdot n_1^T}}
> $$
>
> where $\mathcal{L}_1$ and $\mathcal{L}_2$ are the losses of positive pairs for anchor $m_1$ and $m_2$ in $M$ respectively,
>
> and (for TempCLR)
>
> $$
> \mathcal{L}_{TempCLR} = -\log\frac{e^{m_1 \cdot n_1^T + m_2 \cdot n_2^T}}{e^{m_1 \cdot n_1^T + m_2 \cdot n_2^T} + e^{m_1 \cdot n_2^T + m_2 \cdot n_1^T}}
> $$
>
> where $\mathcal{L}_{TempCLR}$ is the loss for $M$.
>
> Then, we take the positive pair $(m_1,n_1)$ as one example and derive the gradients w.r.t. $m_1 \cdot n_1^T$,
>
> (For VideoCLIP)
>
> $$
> \frac{\partial \mathcal{L}_{VideoCLIP}}{ \partial(m_1 \cdot n_1^T)} = \frac{-e^{m_1 \cdot n_2^T}}{e^{m_1 \cdot n_1^T}+e^{m_1 \cdot n_2^T}}
> $$
>
> and (For TempCLR)
>
> $$
> \frac{\partial \mathcal{L}_{TempCLR}}{ \partial(m_1 \cdot n_1^T)} = \frac{-e^{m_1 \cdot n_2^T}}{e^{m_1 \cdot n_1^T}e^{m_2 \cdot n_2^T - m_2 \cdot n_1^T}+e^{m_1 \cdot n_2^T}}
> $$
>
> The comparison is then
>
> $$
> \frac{\partial \mathcal{L}_{VideoCLIP}}{ \partial (m_1 \cdot n_1^T) }  / \frac{ \partial \mathcal{L} _{TempCLR} }{ \partial (m_1 \cdot n_1^T) }
> = \frac{(e^{m_1 \cdot n_1^T}+e^{m_1 \cdot n_2^T})^{-1}}{(e^{m_1 \cdot n_1^T}e^{m_2 \cdot n_2^T - m_2 \cdot n_1^T}+e^{m_1 \cdot n_2^T})^{-1}}
> $$
>
> As such, we can observe that the optimization of the positive pair in TempCLR $m_1 \cdot n_1^T$ clearly considers the difference when $m_2$ is matched with each element in $N$, \ie, $e^{m_2 \cdot n_2^T - m_2 \cdot n_1^T}$. In other words, the optimization of $m_1 \cdot n_1^T$ also depends on the correlation between $m_2 \cdot n_2^T$ and $m_2 \cdot n_1^T$, which is the temporal context.
>
> Similarly, we take $m_1 \cdot n_2^T$ as one example and compare the optimization of negative pairs.
>
> $$
> \frac{ \partial \mathcal{L}_{VideoCLIP} }{ \partial(m_1 \cdot n_2^T )} / \frac{\partial \mathcal{L} _{TempCLR}}{ \partial (m_1 \cdot n_2^T) } = \frac{(e^{m_1 \cdot n_1^T - m_1 \cdot n_2^T } + 1)^{-1}}{(e^{m_1 \cdot n_1^T - m_1 \cdot n_2^T + m_2 \cdot n_2^T - m_2 \cdot n_1^T} + 1)^{-1}}
> $$
>
> Then, we can also observe that optimization of negative pairs only considers unit-level comparison $m_1 \cdot n_1^T$ and $m_2 \cdot n_2^T$ in VideoCLIP while ours also takes temporal context, i.e., the matching difference $m_2 \cdot n_2^T - m_2 \cdot n_1^T$ into consideration.
>
> We provide detailed calculation of gradients in Supp A.5

---

> > ### Author Response · Authors · 2022-11-19
> > **Response to Reviewer e5YM (2/2)**
> >
> >
> > **Question 2**: Comparison with "Self-supervised Spatiotemporal Learning via Video Clip Order Prediction" (Xu et al.CVPR'19)
> >
> > (*Temporal granularity*) Though Xu et al. has propose the order shuffling before, they do not consider the temporal granularity in shuffling. As discussed in Sec. 5.2, directly applying those shuffling approaches to our network result in sub-optimal performance, which then motivates us to consider the temporal granularity at unit(clip)-level and segment-level and design a design a specific strategy for video-paragraph pretraiing.
> >
> > (*Flexibility in length*) Our approach is flexible to various sequence lengths and is generalizable to new tasks where Xu et al. can only predict order for sequences of fixed length and can hardly model temporal context of the entire video.
> >
> > (*Benefiting comparison \& alignment between two sequence*) The order prediction mechanism in Xu et al. cannot be directly used for aligning units with similar semantic meanings. After all, given two sequence of clips, the classifier will make the same prediction if the orders of clips in each sequence are the same, regardless of the content.
> > However, our approach calculates sequence-level distance and try to align the sequences with the same spatial-temporal meaning. With DTW-releated measure, the shuffling approach is used to break the consistency of temporal dynamics between sequences and generate negative samples which is supposed to have large distance with the anchor. In this way, the model is trained to be aware of the temporal context of the entire sequence and then facilitate the matching between units (i.e., caption-clip pairs).
> >
> > **Question 3**: Comments on writing and notation.
> >
> > Thanks for mentioning this. We have correct the typos and updated the table notations. We have 1) updated the introduction to explain how our approach address the confusing cases where two video clips are visual similar but should be matched with different captions, 2) included new related work to better explain the background. We have also 3) provided a summary for approach explanation in Sec. 3.2, and 4) discussed more about our negative sampling strategies.
> >
> > **Question 4**: Regarding contributions and novelty.
> >
> > There are indeed existing work based on order shuffling, including paper mentioned by the reviewer and a few more literature updated in the paper.
> > However, as acknowledged other reviewers, our approach is sufficiently novel and new under the setting of video-paragraph pre-training for zero-shot transfer. It is also important to formulate the videos (paragraphs) as sequences and compare them directly.
> >
> > (*temporal granularity in shuffling*) Furthermore, we consider the alignment at unit(clip)-level and segment-level, and then propose a new shuffling strategy considering temporal granularity to generate negative samples. From our ablation study, on the other hand, we observed that directly applying the previous approaches to aggressively shuffle all clips will break the temporal succession within each segment and such negative samples will hurt training performance.
> >
> > (*Easy adaptation*) Comparing to related work in multi-modal pre-training, our approach is orthogonal to them as they only focus on unit-level comparison but does not consider the global sequence level comparison. Meanwhile, our approach can be easily combined with those approaches to improve the performance.

---

> > > ### Comment · Reviewer_e5YM · 2022-11-21
> > > **Response to Rebuttal**
> > >
> > > Thanks for the authors' rebuttal. The rebuttal gives deep analysis into the difference between TempCLR and previous training frameworks both theorectically and empirically. Though the techinical novelty is somewhat limited, still having contribution to the area of text video alignment. Thus, I would like to raise my rating to marginally above accepctance.

---

> > > > ### Author Response · Authors · 2022-11-23
> > > > **Thank you for your positive feedback**
> > > >
> > > > We appreciate your positive feedback and acknowledging that our analysis is deep. Regarding technical novelty, as acknowledged by other reviewers, we would like to highlight
> > > >
> > > > 1) that our motivation is sufficiently novel to explicitly utilize the temporal context in aligning video and paragraph by directly performing sequence-level comparison, and
> > > >
> > > > 2) that our approach is novel as it considers both global and local granularity in unit shuffling and analyzing temporal dynamics. Meanwhile, as experimentally demonstrated in Sec. 5.1, our training strategy can also benefit unit-level comparison.
> > > >
> > > > We are happy to provide more explanation if you have any other questions. Thank you very much.

---

### Official Review · Reviewer_kTb8 · 2022-10-24

**Confidence:** 4
**Correctness:** 3
**Technical Novelty And Significance:** 3
**Empirical Novelty And Significance:** 3
**Recommendation:** 6

**Clarity, Quality, Novelty And Reproducibility:**

In my opinion, the motivation of this paper and the problem it addresses are interesting and novel. However, I'm not sure about the generalisation ability of the method as I wrote above and specifically there are a few limitations or aspects of it that were not discussed. Regarding originality, considering temporal information as a pre-text signal for pre-training has been discussed and considered before (in several papers), but AFAIK not in this specific context and setting.

[3] Lee, Hsin-Ying, et al. "Unsupervised representation learning by sorting sequences." Proceedings of the IEEE international conference on computer vision. 2017.
[4] Misra, Ishan, C. Lawrence Zitnick, and Martial Hebert. "Shuffle and learn: unsupervised learning using temporal order verification." European conference on computer vision. Springer, Cham, 2016.
[5] Xu, Dejing, et al. "Self-supervised spatiotemporal learning via video clip order prediction." Proceedings of the IEEE/CVF Conference on Computer Vision and Pattern Recognition. 2019.


**Strength And Weaknesses:**

Strengths:
1. The paper is well written.
2. Considering the clip-level temporal context as aligned to sentence level description of the video is a interesting and novel motivation for learning representation by contrastive learning.

Weaknesses:
EDIT: The rebuttal have addressed the limitations I have written below.

Generally, I agree with the approach of considering a video as a sequence of clips and matching it with a paragraph when it's considered as a sequence of sentences. However, I have a few concerns regarding the way that it's done in the paper and a few limitations/considerations that were not discussed:

1. **Video and clip length**: it is worthwhile to discuss what is the length of a video and a clip in the video in the paper and how that affects the method and the performance and how does the method perform/consider untrimmed vs trimmed videos. For example, the Large Scale Movie Description and Understanding Challenge [1] could fit perfectly in this framework as it is composed of untrimmed Hollywood movies (sometimes an hour plus long) divided into clips, each one associated with a caption. Thumos14 [2] could also fit perfectly as it composed of untrimmed action localization videos. The paper considers benchmarks with much shorter and constrained video, such as YouCook2 and CrossTask where videos are only a few minutes long and the context remains the same and something-something dataset where videos are a few second in length. In my opinions, the generalisation ability of the method to longer and untrimmed videos should be discussed.

2. **Change of context**: There is no discussion on how the change of context affects the method and the learned representation. For example, say that we take a video composed of 2 completely different scenes (as depicted in Thumos and LSMDC for example), how would that affect the performance, given that the method tries to find a matching that produces consistent temporal dynamics where in such example, there is an abrupt change in the temporal dynamics? I want to be clear that I'm not arguing that the method would not work, I argue that given the emphasis on temporal consistent, a case where there is a clear change in the temporal context in a video should be discussed.

3. **Dividing a video into clips**: does the method assumes that the video is already divided into clips? how would the method perform if such a division is not given or it can be easily divided into clips by running shot detection?


[1] https://sites.google.com/site/describingmovies
[2] https://www.crcv.ucf.edu/THUMOS14/home.html

**Summary Of The Paper:**

The paper addresses the problem of video and text matching and pre-training in a contrastive learning contrast. Specially, the paper argues that previous methods considered the "unit-level" context, i.e. : the similarity of one given clip and sentence pair, but did not consider the global context of the clip as part of a longer video or the sentence as part of a longer paragraph. Thus, the paper addresses video and paragraph matching and explicitly considers the temporal alignment of clips and sentences.


To this end, the paper presents a method of using dynamic time warping to match a sequence of sentences (i.e. a paragraph) with a sequence of clips (i.e. a video) along with a temporal contrastive learning framework where negative samples are obtained by shuffling the temporal order of the anchor samples. Thus, the learned representation can facilitate better temporal paragraph/video alignment. The method can also be used for learning video representation without considering video-text pairs. The paper presents experiments on video retrieval, action step localisation and few-shot action recognition.



**Summary Of The Review:**

In my opinion, there are a few aspects and limitations of the method which are not discussed in the paper and affect the applicability of the method to other, less constrained, datasets. Also, the technical novelty is a bit limited as using temporal ordering as a self-supervision signal has been proposed and discussed in various papers before.

---

> ### Author Response · Authors · 2022-11-19
> **Response to Reviewer kTb8 (1/2)**
>
>
> We thank the reviewer for acknowledging our novelty and raising this insightful discussion about the generalization ability of our approach, for which we share our thoughts below.
>
> **Question 1-1**: Discussion on length of clips.
>
> We follow the data processing pipeline in VideoCLIP and evenly split the entire video into clips. VideoCLIP sets a sampling rate of 30FPS and sets a window of one second (i.e., 30 consecutive frames) to build clips and then extract the corresponding S3D features.
> Even though the VideoCLIP model has been trained from S3D features, extracted based on 30 FPS, our TempCLR is still robust to clip features where the clips are defined with other configurations.
>
> For example, when we adjust the sampling rate to 16FPS, we can still achieve similar performance in both zero-shot action step localization (recall 36.9\% for 30FPS vs. 37.1\% for 16FPS) and zero-shot full video retrieval (R@1 83.5\% for 30FPS Vs. 83.2\% for 16FPS).
> In addition, when the window size is changed, e.g., a clip consists of frames of two seconds, we can also perform bi-linear interpolation to estimate features for each second.
>
> **Question 1-2**: Discussion on video length and application on trimmed \& untrimmed videos.
>
> (*Clarification of assumption*) Our approach is based on the assumption that the video content consists of successive steps/stages for a common concept (e.g., event or action). For example, instructional videos consist of multiple steps where different steps are correlated. Even a simple action, "playing the baseball" in a trimmed video, can still be further segmented into successive fine-grained components. In this way, we can then utilize the consistency of temporal dynamics between two sequences (i.e., video and its paragraph) and apply TempCLR.
>
> (*Trimmed*) Thus, assuming that the content in one trimmed video is only for one concept, e.g., few-shot action recognition, our approach can be applied and has provided clear performance gain. Regardless of the video length, we believe our approach can be naturally generalized.
>
> (*Untrimmed Video of one concept*) We then consider the scenario where unrelated content, such as background, exists between successive steps. Similar to the full-video (background kept) retrieval, as the features of all captions are distant from the video clips of background, with DTW-related measures (e.g., DTW, OTAM, and other variants), we can still retrieve the paired videos correctly. We note that the VideoCLIP has been pre-trained to compare clip captions across videos, and the features have already been discriminative. However, we can still combine "up-paired" and "seg-unit" as the negative sampling strategy when TempCLR is trained from scratch.
>
> (*Untrimmed Video with multiple concepts*) Finally, we consider the scenario where multiple concepts exist in one video. Since the concepts are not guaranteed to be successive or even related, we thus need to prepossess the entire videos and split them according to the concepts. Then, TempCLR can be applied to each segment whose content is successive. However, if the concepts are still related, we think our approach can be generalized with minor modifications in data sampling.
>
> In both Thumos and LSMDC, we need first to split the untrimmed video if the concepts are not successive. However, we note that the length of each split video part can still be relatively long (e.g., we can split a two-hour video into different parts, and each part can be around 20 mins.). For example, tools relating to action localization or shot detection can be utilized. In addition, for LSMDC, the scripts can also be a good indicator. We acknowledge that split videos at distant periods can be the same concept. However, we think it is safer to apply TempCLR on each separately and then study the application in their combination. We leave this for future work and are happy to explore this scenario.

---

> > ### Author Response · Authors · 2022-11-19
> > **Response to Reviewer kTb8 (2/2)**
> >
> > **Question 2**: Discussion of context change
> >
> > (*Approach Explanation*) We would like to re-highlight that TempCLR is based on the assumption that the video content consists of successive steps/status for a common concept such that the consistency of temporal dynamics can then be discussed. When the scene changes while the succession across status is kept, e.g., two persons are discussing, and the scenes for them are different ([example](https://sites.google.com/site/describingmovies/previous-years/lsmdc-2019)  shown in LSMDC), we think our approach can still be applied.
> >
> > (*Scene change with the same semantic meaning*) As shown in Fig.8 in Supp, for one caption, "add oil to the wok", we replace its clips in the original video with the clips from another video, where the replaced ones also talk about the caption to synthesize a new video. Then, we use DTW to align the newly generated video with the paragraph, and our model can still perform the alignment properly.
> >
> > (*Scene change with background*) As shown in Fig. 9 in Supp, we choose to manipulate the video by inserting a video segment of the restaurant (background) into the video. Though the scene changes, as the features in the common space are robust to the background, we can still make the good alignment.
> >
> > (*Scene change with different semantic meaning*) However, when the change of scenes breaks the succession, we should apply related approaches to pre-process the entire video. We can perform shot detection or follow the idea proposed in [6] to predict the activity score for each clip and apply a threshold to split the full video into the different parts and then apply TempCLR on each part individually. Similarly, language tools can also be used if the scripts are given.
> >
> > (*Comparison with related work*) We have checked the [presentation](https://drive.google.com/file/d/1eMLMo4SlEIZhoLXnT-aG3w6D4M45Wisx/view) of the winner in LSMDC challenge 2019. We found their approach consists of behavior consistency between captions and clips (unit-level comparison) in the task of characters filling in. Thus, we can adapt our approach to also model the dynamics of video clips in the full temporal context and find the option of filled characters which can result in similar dynamics in the feature space.
> >
> > Overall, we believe the modeling of successive steps in each concept is important, and our approach can be robust and generalized to different scenarios. We believe the change of context would an important aspect of temporal modeling, and we are happy to leave it for future work.
> >
> >
> > **Question 3**: Clip division
> >
> > We follow the clip-dividing strategy in VideoCLIP, i.e., one-second window at 30 FPS, and do not hold any assumption on clip division.
> > Even for the split of the segment, in the full-video retrieval, when the background is kept in the full video, we do not need the annotation of the segment to recognize which part is the background. Instead, we feed all of the clips into the Transformer architecture and thus do not have any assumptions.
> > However, as discussed above, we think using shot detection to prepossess the video and filter out clips of background can be beneficial.
> >
> > **Question 4**: Comparison with Related Work (Regarding originality)
> >
> > (*Flexibility in video length*): We thank the reviewer for bringing our attention to the related work. Though the ideas of shuffling has been studied before, including [3-5] listed by the reviewer, the shuffling mechanism in TempCLR is more flexible and can be easily generalized to sequences with various length. However, for the related work, the length of the sequence has been fixed and too small (e.g., no more than 4) to model the temporal context of the entire video.
> >
> > (*Facilitating alignment*): Secondly, the order shuffling in [3-5] is used as augmentation for the order-prediction task, while it does not clearly help with the alignment between sequences. For two sequences where the shuffling orders within each sequence are the same, the classifier will have the same ground truth, regardless of the content (e.g., swimming and cooking).
> >
> > (*Shuffling with temporal granularity*) Finally, our approach is applied to a multi-modal domain and considers temporal granularity. However, the previous work about order shuffling only considers random shuffling, and the difficulty of shuffled videos is not hard. We also provide a detailed discussion in Sec 5.2 and demonstrate directly that the previous shuffling approach results in sub-optimal performance.
> >
> > [3] "Unsupervised representation learning by sorting sequences."
> >
> > [4]  "Shuffle and learn: unsupervised learning using temporal order verification."
> >
> > [5] "Self-supervised spatiotemporal learning via video clip order prediction."
> >
> > [6] AutoLoc: Weakly-supervised Temporal Action Localization in Untrimmed Videos (ECCV'18)

---

> > > ### Comment · Reviewer_kTb8 · 2022-11-22
> > > **Response to rebuttal**
> > >
> > > I would like to thank the authors for the detailed rebuttal and for addressing my concerns regarding the limitations of the work and its applicability to other scenarios of untrimmed videos with possible change of context. I acknowledge that such scenarios are a bit outside the scope of the paper and I appreciate for the discussion in the rebuttal on how the method might still be applied. As those concerns have been mitigated, I have updated the score given to 6, marginally above the acceptance threshold.

---

> > > > ### Author Response · Authors · 2022-11-23
> > > > **Thank you for your reply**
> > > >
> > > > We appreciate your positive feedback, and we believe the discussion required by you are important for future development. We are happy to provide more explanation if you have any other questions. Thank you very much.

---

### Official Review · Reviewer_pEeR · 2022-10-25

**Confidence:** 3
**Correctness:** 3
**Technical Novelty And Significance:** 3
**Empirical Novelty And Significance:** 3
**Recommendation:** 6

**Clarity, Quality, Novelty And Reproducibility:**

The writing quality and clarity of the paper is good, though there are a few implementation details I might have missed (see questions above). I think the method is sufficiently novel, as it differs in may ways from prior works that inspired this work.

**Strength And Weaknesses:**

## Pros
- The proposed method of matching sequences for contrastive learning is instinct, and the idea of shuffling sequence order to create negatives also arises naturally from the motivation of using global context of long videos.
- Experiments are conducted on three different downstream tasks, under zero-shot, few-shot and fine-tuning settings, with significant improvements over the VideoCLIP baseline.
- I find the ablation studies and visualizations quite helpful in understanding the mechanism of individual components in the approach.
- The paper is very well written, with concise problem formulation, and intuitive figures to illustrate the idea.

## Cons
- While not directly comparable, the paper can benefit from a discussion of prior work using shuffled clips for video unsupervised learning ([Shuffle & learn](https://arxiv.org/abs/1603.08561,), [OPN](https://arxiv.org/abs/1708.01246), [VCP](https://arxiv.org/abs/2001.00294) etc.).
- It is interesting that incorporating negative samples from other instances ("unpaired" in table 7) performs the worst, as instance discrimination is typically helpful in video contrastive learning. I wonder if it is possible to use both unpaired videos and shuffled videos as negatives in contrastive learning?
- From supplemental table 8 using VideoCLIP with `DTW + Cap. Avg.` metric seems to already produce strong results. Does this mean the TempCLR pretraining is not essential to achieving good retrieval performance? Is the same observation true for other tasks too?
- For text-to-video retrieval, including results on more datasets (DiDeMo, ActivityNet Captions etc.) should make the comparisons more convincing.

## Questions
- In table 1, is there a particular reason why cap. avg. measure is not reported, or cannot be used for full video retrieval with background?
- Can the authors comment on the number of negatives per video used in TempCLR, as well as the robustness of the method to the size of negative set?

**Summary Of The Paper:**

This work studies video-text contrastive learning with the focus on learning the global temporal context over long videos. The authors propose a new contrastive learning method, TempCLR, which poses the pairwise distance as matching cost between text (sentences) and visual (clip) sequences, computed by dynamic time warping. Negative samples are constructed by shuffling the clip order of a positive video, as well as frame order within each clip. Experiments are conducted on zero-shot video retrieval, action step localization, and few-shot action classification tasks, where TempCLR improves the results over VideoCLIP and other baselines, especially when the task requires global temporal information.

**Summary Of The Review:**

Given the simplicity and effective of the proposed method, I lean towards accepting the paper, though there are a few questions on the constructions on the negative samples, among other minor concerns listed above. I look forward to the authors' response to further clarify these issues.

---

> ### Author Response · Authors · 2022-11-19
> **Response to Reviewer pEeR (1/3)**
>
> We thank the reviewer for acknowledging our approach is sufficiently novel. The reviewer asks questions about comparison with related literature and clarification of implementation, for which we have provided our answers below.
>
> **Question 1**: Comparison with existing video unsupervised learning works.
>
> We thank the reviewer for bringing our attention to related work, and our TempCLR differs from them in three aspects.
>
> (*Flexibility in video length*) Though the idea of shuffling clips in video unsupervised learning has been studied before, the mechanism of shuffling is not flexible. For example, OPN and Shuffle \& Learn only trains the classifier to predict the order among a fixed number of frames (4 in OPN and 3 in Shuffle \& Learn), while TempCLR utilizes the DTW to compare the sequences with various length. In addition, the classifier for order prediction may suffer from position bias and may not treat all features equally.
>
> (*Facilitating alignment*)
> Secondly, the previous work cannot be directly used to facilitate the alignment between two sequences.
> For example, for sequences with completely different semantic meaning, they may have the same shuffling order prediction. Though the models has been trained to predict the temporal shuffling types such as VCP (in fact, the spatial augmentation is also treated as classification categories), the model is still not explicitly trained for sequence alignment. In contrast, for our approach, we utilize the mechanism in DTW measure to indicate the global temporal context constraint where the distance should be high if two sequences are not semantical-related. Also, the motivation of our approach is to utilize temporal shuffling to break the consistency of temporal dynamics between two sequences such that the most confusing case of local matching can be automatically detected.
>
> (*New domain \& setup, design of shuffling strategy*)
> Previous approaches are developed for video-only self-supervision while the setup in our approaches is about video-paragraph pretraining. Also, the duration of video is longer and more challenging. In addition, the temporal granularity, i.e., unit(clip)-level and segment-level, should be considered. Our discussion on Sec. 5.3 also shows that directly shuffling all clips to break the succession within each segment may result in sub-optimal performance in zero-shot adaptation. Furthermore, similar to how we apply our approach to VideoCLIP to build TempCLR, we can also apply it to other framework.
>
> We have updated them in the related work (Sec. 2).
>
> **Question 2**: Clarification on Table 7 & Performance by combining "unpaired" with "shuffled" videos.
>
> As Table 7 conducts an ablation study on the negative sampling strategy on Cross-Task under a supervised setting, we would like to remind the reviewers to compare them with the performance of VideoCLIP baseline 47.3\%, which has been reported in Table 4 (right).
>
> (*Bias in VideoCLIP initialization*) The "unpaired" generate negative samples from unpaired videos. However, as the pre-trained VideoCLIP model has already been learned from comparing clips and captions from different videos and paragraphs, the new knowledge to be learned by employing "unpaired" can be limited.
> Instead, the pre-trained VideoCLIP model does not model global temporal context; generating negative samples by shuffling the order of units in the positive sequence in TempCLR can help with generalization to new tasks clearly.
>
> (*Joint training strategy*) We then combine "unpaired" and "seg-unit", but the performance is still 52.5\% compared with 52.5\% by "seg-unit". As mentioned above, the ability regarding instance discrimination has been fully trained in the VideoCLIP model and the modeling of global temporal context is more important in our TempCLR. However, we believe that both instance discrimination and the global context of correlation are essential for sequence alignment. When the model is trained from scratch, such a joint training strategy is necessary.
> We have updated the discussion in Sec. 5.2.

---

> > ### Author Response · Authors · 2022-11-19
> > **Response to Reviewer pEeR (2/3)**
> >
> > **Question 3**: Strong results on Video Retrieval
> >
> > Ensembling metrics is indeed good for global retrieval (video-level). However, it is hard to combine Cap. Avg. and DTW-related measure for fine-grained retrieval (segment/moment-level), e.g., for action step localization on CrossTask, we cannot directly implement this metric ensembling technique. Instead, implementing TempCLR pre-training can improve the performance against the VideoCLIP baseline on both zero-shot and supervised setups.
> >
> > Meanwhile, for zero-shot transfer, we think the domain gap between the pre-train set and the downstream set should be considered. We have conducted extra experiments on DiDeMo in the Sec. A.4.2. However, even with metric ensembling (i.e., OTAM + Cap. Avg.), we observe that the performance (R@1) in full-video retrieval is only ~10.6\% (background removed) and 9.0\% (background kept) by VideoCLIP. In contrast, applying TempCLR can improve performance consistently.
> >
> > Finally, we believe this metric ensembling can be generalized to other tasks, such as few-shot action recognition.
> >
> >
> > **Question 4**: More Experiments.
> >
> > As requested, due to the time limitation, we have conducted extra experiments on DiDeMo. We evaluated it under both clip retrieval and full video retrieval setup. For clip-caption retrieval, we follow the commonly used metrics (R@1, R@5) as metrics and achieve a consistent performance improvement, i.e., from (16.39\%, 47.18\%) by VideoCLIP to (17.7\%, 48.02\%) by TempCLR. For full video retrieval, with OTAM as a measure, the performance (R@1) is also improved from 8.9\% to 10.4\% (background removed), and 8.1\% to 9.3\% (background kept). We explained the implementation details and included the full results in Sec. A.4.2 in Appendix.
> >
> >
> > **Question 5**: Cap. avg. measure on full video retrieval (background kept) with the background.
> >
> > We thank the reviewer for mentioning this, and we have updated them in Sec. A.4.3 in the appendix.
> >
> > (*Performance with Cap. Avg. as measure*) As VideoCLIP has been well-trained on the full HowTo100M dataset to compare the clips and captions, the model may benefit from the measure Cap. Avg. Specifically with Cap. Avg. as the measure, though the performance of R@1 drops from 73.6\% by VideoCLIP to 71.7\% by TempCLR, the performance of R@5 is improved from 94.1\% by VideoCLIP to 94.5\% by TempCLR. As the model VideoCLIP has been pretrained to discriminate background and clips from unpaired videos, with Cap. Avg as the measure, the features of captions can still be distant from features of background clips, and the features in common space of VideoCLIP or TempCLR can still be robust to the background and maintain high recall.
> >
> > (*OTAM Vs. DTW*) Regarding performance by TempCLR with DTW-related measures, we note that DTW has a strong boundary assumption, and thus, the performance is limited. However, OTAM can avoid the assumption and can then improve R@1 of TempCLR to 72.2\%. Meanwhile, we believe necessary modifications on DTW-related measures can be further developed to improve the training efficiency, and we leave this for future work.
> >
> > (*Temporal Context is still necessary*) We believe in using Cap. Avg is not enough. Even though VideoCLIP has achieved strong R@1 with Cap. Avg. as a measure, it drops to 56.7\% when OTAM is used as a measure, and the performance gain by ensembling Cap. Avg with OTAM (DTW) is limited, i.e., 74.5\% (73.8\%).
> > In contrast, though our TempCLR is only trained on a subset (7.5\%) of full train set, it can better explore the temporal context and can reach 77.5\% (76.7\%) when OTAM (DTW) is ensembled with Cap. Avg., which clearly demonstrates the importance of modeling temporal context.
> >
> > (*Impact of domain gap*) Meanwhile, we would like to note that, as the domains of HowTo100M and YouCookII are similar, the model may thus achieve good performance on YouCookII. However, the zero-shot transfer performance on DiDeMo is not as high, e.g., the full video retrieval by VideoCLIP with Cap. Avg. as the measure is only about 8.9\% for background removed and 7.1\% for background kept. In contrast, using DTW-related measures can consistently outperform the performance when Cap. Avg. is used as a measure, i.e., 10.4\% for background removed and 9.3\% for background kept.

---

> > > ### Author Response · Authors · 2022-11-19
> > > **Response to Reviewer pEeR (3/3)**
> > >
> > > **Question 6**: Robustness to negative set
> > >
> > > We adjust the size of negative samples from (16, 32, 64) and use the zero-shot performance on action step localization and full-video retrieval for comparison. We use Recall as metric.
> > > For zero-shot action step localization, the performance are (36.5, 36.9, 37.1); for zero-shot video retrieval, the performance is (82.9, 83.5, 83.5).
> > >
> > > By increasing the negative size from 16 to 32, we can see the performance can be improved slightly. However, when we keep increasing the size to 64, there is no significant gain. To note, for the shuffling strategy, shuffling at the segment-level is important. Also, during pre-training, as we directly use the data sampling strategy in VideoCLIP, the number of segments existing in each video is only 6 on average.
> > > To reach reasonably high performance but still train the network efficiently, we set the size of negative samples as 32 for all experiments. We have updated them in Sec. A.5 in Appendix.

---

> > > > ### Comment · Reviewer_pEeR · 2022-11-23
> > > > **Thank you!**
> > > >
> > > > I appreciate the elaborated response from the authors, which has clarified my initial questions including the relation to prior works and the selection of negative examples. Though it seems that the advantage of proposed method for video retrieval is sometimes marginal compared to clip-level matching baseline, I agree with authors that strong temporal alignment has a greater value in finer-grained tasks like action localization, which requires reasoning beyond individual video clips. I am inclined to keep my initial rating after reading other reviews and authors' responses.

---

> > > > > ### Author Response · Authors · 2022-11-23
> > > > > **Thank you for your reply.**
> > > > >
> > > > > We appreciate your positive feedback and would be happy to address any more questions you have. Thank you again very much.

---

### Official Review · Reviewer_KVqq · 2022-10-26

**Confidence:** 4
**Correctness:** 4
**Technical Novelty And Significance:** 3
**Empirical Novelty And Significance:** 3
**Recommendation:** 6

**Clarity, Quality, Novelty And Reproducibility:**

Exploiting temporal dynamics to align videos and paragraphs through global and local granularities is novel. This may provides insights for studying the temporal characteristics of videos.

**Strength And Weaknesses:**

Strengths:
1. The overall idea is simple and proved to be effective empirically.
2. The core idea of exploiting temporal dynamics to align videos and paragraphs through global and local granularities provides insights for studying the temporal characteristics of videos.
3. Extensive experiments have been conducted to verify the effectiveness of the proposed sampling strategy and DTW-based metric.
4. The proposed method achieves good results on three tasks.

Weaknesses&Questions:

Though the overall paper is interesting, I have the following concerns:

1. The authors claimed that the motivation is to address the case where two clips are visually similar but are from different segments. However, it seems this problem has not been well addressed through the proposed scheme.

2. I suggest the authors provide more discussions on the difference between the proposed method and other works that align video and text, such as [a].

3. The performance of the zero-shot action step localization task is not that good compared with other methods (see Table4).

4. The loss L_{seq} brings improvements on most of the settings while leading to worse results on Action Step Localisation (compared with Video Clip in Table 6(b)). Could the authors provide some discussions?


[a] Temporal Alignment Networks for Long-term Video. CVPR2022


**Summary Of The Paper:**

This paper introduces a contrastive learning-based method for learning video representation. The core idea is to use the temporal order of video segments and sentences to align the temporal-sensitive representation. Meanwhile, a negative sampling strategy based on temporal granularity and shuffling the units in the positive sequence is proposed to obtain positive and negative samples. Experimental results on three tasks show the effectiveness of the proposed method.

**Summary Of The Review:**

The overall idea is simple but effective and achieves consistent improvements on three tasks, including video retrieval, action step localization, and few-shot action recognition.

---

> ### Author Response · Authors · 2022-11-19
> **Response to Reviewer KVqq (1/2)**
>
> We thank reviewer for acknowledging the novelty and our contribution for providing insights for studying temporal characters of videos.
> The reviewer asks about how our approach solves motivation problems, discussion with related work, and more experiment analysis, for which we provide our thoughts below.
>
> **Question 1**: How does TempCLR address the *confusing case* where two clips are visually similar but from different segments.
>
> A1: We address the confusing case by two steps.
>
> (*Step 1: Employing DTW as measure*) First, we employ DTW-based measure which can consider the temporal context in matching video clips and captions. Under the constraint of temporal order, DTW calculates an optimal alignment between the sequence of clips and the sequence of sentences/captions. Each clip is then matched with one sentence.
>
> (*Step 2: Obtain context-aware representation*) Previous approaches, such as VideoCLIP, only perform unit-level comparisons and do not consider sequence-level comparisons during training. In contrast, we propose TempCLR to learn to extract features and compare the sequences with different orders. In this way, the features are trained to be aware of the temporal context of the full video and can thus use DTW to obtain clip-caption matchings, which can distinguish visually similar clips in confusing cases.
>
> (*Experiments*) As compared in Table.6(a), when we align the video and its paired paragraph through DTW, our TempCLR can correctly match 91.6\% of caption-clip pairs from the derived alignment. The visualization in Fig. 3 shows one detailed example. In Table 3, even without DTW measures, the extracted features can better facilitate the caption-clip retrieval task. In addition, we conduct extra video-retrieval experiments on another dataset DiDeMo (Sec. A.4.2), which shows that DTW can help improve the video retrieval performance consistently, in particular when the background is kept. All of the experiments mentioned above do demonstrate the robustness of TempCLR in addressing confusion cases.
>
> We have also updated the explanation in the introduction part.
>
> **Question 2**: Comparison with Temporal Alignment Networks for Long-term Video (CVPR'22) [a].
>
> We thank the reviewer for bringing our attention to the related work. Compared with [a], TempCLR focuses more on obtaining features that are aware of temporal context and is also orthogonal to [a].
>
> First, the training objectives in [a], i.e., alignability loss and temporal correspondence loss, are still based on caption-clip (unit-level) comparison.
> Secondly, [a] proposes a co-training strategy and uses the mutual agreement, between a vision-only encoder and a vision-language encoder, to denoise the noisy annotation. For our TempCLR, we follow the data sampling strategy in VideoCLIP to mitigate the issue caused by noisy annotation and then explicitly perform a sequence-level alignment. Thus, TempCLR can be adapted in [a] intuitively.
>
> We have updated the comparison in Sec. 2.
>
> **Question 3**: Regarding zero-shot action step localization task
>
> As what we have explained in the Table 4 caption, due to the limitation of computational resources, we only use a subset (i.e., 7.5\% of the full HowTo100M data) and train the model from VideoCLIP. Meanwhile, finetuning from VideoCLIP with limited data has an overfitting risk and the recall of zero-shot action step localization drops to 33.5\% after finetuning with the loss in VideoCLIP.
>
> However, our TempCLR can still improve the recall to 36.9\%. Meanwhile, our approach can be adapted to different baselines. We believe the effect of TempCLR can be better explored when the model is trained from scratch or from a stronger baseline such as MIL-NCE.
>
> In addition, we break the performance according to the task type and observe our approach can provide clear gain in recall (42.7\% by TempCLR Vs. 38.1\% by VideoCLIP) on 9 tasks which requires sophisticated steps for food making (e.g., Make Meringue) and may emphasize the importance of temporal context in alignment as well as benefit from our approach.

---

> > ### Author Response · Authors · 2022-11-19
> > **Response to Reviewer KVqq (2/2)**
> >
> > **Question 4**: Discussion on Table 6(b) regarding performance drop in action step localization.
> >
> > A4: We assume the reviewer refers to the performance difference between 52.5% and 52.3% in Table 6(b). Though we believe these two performances are still comparable, we think the reason for this slight drop is caused by the gap of temporal pattern between HowTo100M and CrossTask.
> >
> > In Table 6(b), we perform an ablation study on action step localization in a supervised setup on the CrossTask dataset.
> > Then, TempCLR is trained with a subset (7.5\%) of HowTo100M and may overfit the temporal pattern exhibited in the subset, while VideoCLIP may not.
> > As the training data of CrossTask is also limited (540 videos), when TempCLR is used as initialization, the model may find it hard to overcome the gap of the temporal pattern. However, when VideoCLIP is used as initialization, the model may fully exploit the temporal pattern exhibited in the CrossTask train set and thus have higher test performance.
> >
> > However, as discussed in the second paragraph under Sec. 5.3 before, we think the two scores are still comparable. Furthermore, with the model pretrained by either VideoCLIP or TempCLR, when the model is adapted to CrossTask, comparing with VideoCLIP baseline 47.3\%, including $\mathcal{L}_{seq}$ can consistently provide performance gain.
> > We have completed the discussion in the Sec. 5.3.

---

> ### Author Response · Authors · 2022-12-07
> **Further Discussion**
>
> We would like to thank the reviewer again for providing positive feedback and detailed comments on our paper.
>
> We provided a detailed response to all the raised questions on Nov 18. As the discussion period is ending soon, we would like to check back and see if there are any remaining questions or concerns. We'd be happy to engage in further discussions with the reviewer.

---

### Author Response · Authors · 2022-11-23
**General Response to All Reviewers**

We thank all reviewers for their valuable comments and all positive feedback. We appreciate the reviewer acknowledging the technical novelty. In particular, it is acknowledged by reviewers that our approach is sufficiently novel and can bring new insights to the related domains, and that our motivation is also novel.

Since most reviewers ask for the comparison with related work, besides addressing the concerns of each reviewer separately, we also provide a general summarization below. Compared with the previous work [1-5], our approach differs from theirs in the following aspects.

(*Sequence-level Vs. Unit-level*) Most importantly, we choose to perform the sequence-level comparison directly, which motivates the idea of order shuffling. Different from VideoCLIP and [1], which only perform unit-level (i.e., clip-caption) comparison, we propose to directly perform sequence-level (i.e., video-paragraph) comparison with DTW-related measure. Furthermore, the sequence-level comparison can also benefit the unit-level comparison, which has been both quantitatively and qualitatively discussed in Sec. 5.1.

(*Facilitating alignment*) Compared with [2-5], the shuffling during training aims to facilitate the sequence alignment. In this way, for each anchor sequence, we shuffle the positive sequence to break the temporal consistency while the negative sequence should be distant from the anchor. In contrast, the order shuffling approaches [2-5] only train a classifier to predict the shuffling order, which cannot be used to measure or indicate the semantic correlation between two sequences.

(*Temporal Granularity*) To generate negative samples which can facilitate the network training, we propose to consider both unit(clip)-level and segment-level shuffling. Our discussion on Sec. 5.3 also shows that applying the previous approach by directly shuffling all clips aggressively will break the succession within each segment and result in sub-optimal performance.

(*Flexibility & Adaptation*) Our approach is flexible to various sequence lengths. It is generalizable to new tasks where [2-5] can only predict order for sequences of fixed length (e.g., 4), which is too small to model the temporal context of the entire video. Furthermore, our approach can be easily applied and combined with other training frameworks.

[1] Temporal Alignment Networks for Long-term Video. CVPR2022

[2] Video Cloze Procedure for Self-Supervised Spatio-Temporal Learning. AAAI 2020

[3] Unsupervised representation learning by sorting sequences. ICCV 2017.

[4] Shuffle and learn: unsupervised learning using temporal order verification. ECCV 2016

[5] Self-supervised spatiotemporal learning via video clip order prediction. CVPR 2019.

---

### Decision · Program_Chairs · 2023-01-20

**Decision:**

Accept: poster

**Justification For Why Not Higher Score:**

issues of clarity

**Justification For Why Not Lower Score:**

all four reviewers are borderline positive but appreciate the ideas presented

**Metareview: Summary, Strengths And Weaknesses:**

Paper proposes a contrastive learning-based method to learn video representations.  The contrastive learning scheme is built on top of the segment temporal ordering and and sentences for positive pairs and shuffled frames and clips for negative pairs.

strengths:
+ good clarity and presentation
+ strong experimentation, analysis and results

weakness:
- some open points and clarifications from the reviewers.  Most seems to have been addressed in the rebuttal.  The authors are strongly recommended to revise their manuscripts accordingly

**Note From Pc:**

if the above contains the word "oral" or "spotlight" please see: "oral" presentation means -> notable-top-5% and "spotlight" means -> notable-top-25%. As stated in our emails, we are disassociating presentation type from AC recommendations